# Self–Similarity Priors: Neural Collages as Differentiable Fractal Representations

**Michael Poli**\*
Stanford University
DiffeqML

**Winnie Xu**\*
University of Toronto

**Stefano Massaroli**\*
Mila
DiffeqML

**Chenlin Meng**
Stanford University

**Kuno Kim**
Stanford University

**Stefano Ermon**
Stanford University
CZ Biohub

## Abstract

Many patterns in nature exhibit *self-similarity*: they can be compactly described via self-referential transformations. Said patterns commonly appear in natural and artificial objects, such as molecules, shorelines, galaxies and even images. In this work, we investigate the role of learning in the automated discovery of self-similarity and in its utilization for downstream tasks. To this end, we design a novel class of implicit neural operators, Neural `Collages`, which (1) represent data as the parameters of a self-referential, structured transformation, and (2) employ hypernetworks to amortize the cost of finding these parameters to a single forward pass. We detail how to leverage the representations produced by Neural `Collages` in various tasks, including data compression and generation. Neural `Collage` image compressors are orders of magnitude faster than other self–similarity–based algorithms during encoding and offer compression rates competitive with implicit methods. Finally, we showcase applications of Neural `Collages` in fractal art and as deep generative models.

## 1  Introduction

*Given a specified image, can one come up with a dynamical system with it as its attractor? (Welstead, 1999)*

Scientific fields are underpinned by a search for structure. Geometry, sparsity and invariances, when appropriately introduced in a mechanistic model, allow us to concisely describe phenomena. To this end, machine learning has been introduced as a means to fix partial priors in a model, and discover the rest through data (Rackauckas et al., 2020; Dao et al., 2020; Bronstein et al., 2021). In general, the notion of structure is also essential for compression: through a suitable choice of language, one can explain phenomena in fewer symbols, yielding shorter representations of the observables (Tishby et al., 2000; Lee et al., 2007). Objects exhibiting *self-similarity* structure are composed of patterns that appear similar to themselves at multiple scales, as shown in Figure 1. This type of structure frequently appears in nature, at different degrees: shorelines, molecules, plants, turbulent flows and basins of attraction of dynamical systems all display elements of self-similarity (Mandelbrot and Mandelbrot, 1982; Song et al., 2005; Vulpiani et al., 2009; Barnsley, 2014). In this work, we explore the role of learning in the automatic discovery of *self-similarity* structure in data, and how it can serve as an inductive bias in machine learning models.

The mathematical embodiment of this idea is found in fractal patterns, which often arise by characterizing limit sets of nonlinear maps e.g. iterations of complex numbers for Julia and Mandelbrot

---

\*Equal contribution authors. Contact email: `poli@stanford.edu`

sets (Julia, 1918; Mandelbrot, 1980). Fractals are scale-invariant: they can be "zoomed in" by increasing the resolution of the limit sets, and manifest arbitrarily similar patterns at different scales. Despite their apparent infinite complexity, a fractal can be uniquely and compactly described by its generating nonlinear map.

A method to discover self-similar structure in data (not necessarily of fractal nature) can then be formalized as an optimization problem: after choosing an appropriate class of contractive, parameterized maps, one searches for parameters such that a given data point can be (approximately) recovered as the (unique) fixed-point of the chosen map. This approach, pioneered in (Barnsley et al., 1986), paved the way for one of the most successful algorithmic applications of self-similarity, *fractal image compression* (Jacquin et al., 1992; Jacquin, 1993; Barnsley et al., 1996; Welstead, 1999; Fisher, 2012). First, an encoding step carries out a search to solve the inverse problem of data to operator parameters via extensive search, or by restricting the class of operators such that a

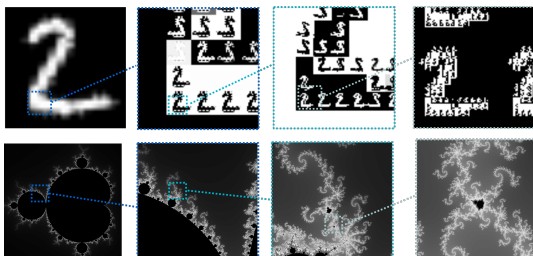

Figure 1: **[Top]** MNIST digit *fractalized* via a `Collage`. The image is represented as the coefficients of a `Collage`, and decoded as its attractor. Magnification is done by decoding at higher resolutions. **[Bottom]** The Mandelbrot set (Mandelbrot, 1980), an example of a fractal displaying self-similarity.

closed-form solution may be found. Then, given the parameters, the decoding step solves for the fixed-point of the operator, corresponding to a corrupted version of the original data. The quality of decoded images i.e. the *loss* of the fractal compression method is directly tied with the expressivity of the class of operators considered, which are often designed to seek self-similarities in pixel space. Yet, larger classes induce more challenging optimization problems, leading to long encoding times.

Here, we propose a novel learning-based technique to extract and utilize self-similar representations of data. We develop Neural `Collages`, a family of differentiable, parametrized operators structured to capture self-similarity between partitions of data. The inverse problem of Neural `Collages` is solved with a single forward pass of a hypernetwork (Ha et al., 2016) trained to generate a set of parameters as the fractal code. This amortized approach is orders of magnitude faster than traditional search based methods. `Collage` operators are composable with neural network architectures and have wide applicability beyond compression, as they can be optimized end-to-end for a variety of tasks including generative modeling.

In data compression, Neural `Collages` preserve advantages of fractal compression methods, with up to $10\times$ (accounting for training time) and $100\times$ (at test time) speedups during encoding. Further, we investigate deep generative models based on Neural `Collages`, where `Collage` parameters assume the role of latent variables of a hierarchical *variational autoencoder* (VAE) (Kingma and Welling, 2013). `Collage` VAEs can sample at resolutions unseen during training by decoding at higher resolutions through their `Collage` operator, revealing additional detail over upsampling via interpolation. Finally, we showcase applications for fractal art, where an image can be "fractalized" i.e. reconstructed as a collage of smaller copies of itself at different scales (see Figure 1, top).

## 2 Problem Setting

We start with an example related to a core idea behind fractal compression and Neural Collages: representing data as the parameters of a function that converges to it.

### 2.1 Representing Data as a Fixed-Point of an Iterated Function

Consider a value $x$ that happens to be of interest for a downstream task, for example regression. Rather than performing computation on it directly, we can use the parameters $\omega$ of a contractive map with $x$ as its fixed-point, for example a simple affine function with parameters $\omega := (a, b)$, $a \in \mathbb{R}, b \in \mathbb{R}$:

$$x_{k+1} = ax_k + b, \quad |a| < 1$$

Given $\omega$, the value $x$ can be recovered by repeatedly applying the map starting from any other initial condition, that is

$$x^* : x^* = ax^* + b.$$

Parameters $\omega$ can then replace $x$ as an alternative representation, using two values $(a, b)$ instead of $x$.

Other classes of functions, if chosen appropriately, can represent data with *less* parameters than its dimensionality. In general, encoding through a function is a lossy process: the original data is recovered up to a degree of accuracy. The insight behind fractal compression and Neural `Collages` is to view data as *sets*, and define iterated functions as ensembles of simple scalar linear functions acting on different subsets. Below, we introduce background required to make these notions precise.

## 2.2 Fractal Compression Background

We proceed by formalizing *fractal data encoding* as an optimization problem (Barnsley and Demko, 1985; Fisher, 2012; Barnsley, 2014).

Let $(\mathbb{X}, d)$ be a complete metric space and $(\mathcal{H}(\mathbb{X}), d_{\mathcal{H}})$ its corresponding Haussdorff metric space, i.e. $\mathcal{H}(\mathbb{X}) = \{\mathbb{A} \subset \mathbb{X} : \mathbb{A} \text{ is compact}\}$. We represent a data point as a set $\mathbb{S} \in \mathcal{H}(\mathbb{X})$. This choice of space supports an data modality-agnostic treatment of fractal data encoding. A concrete realization is discussed for image domains in Section 3. Results on metric spaces are provided as reference in Appendix A.

**Example 1.** *Consider binary images on a square domain. Then, $\mathbb{X}$ is a finite compact subset of $\mathbb{R}^2$. An image $\mathbb{S}$ is the finite set of coordinates of either black or white pixels. Further, each image corresponds to a point in $\mathcal{H}(\mathbb{X})$.*

Let $\{f_1, f_2, \ldots, f_K\}$ be a collection of maps on $\mathbb{X}$, $f_k : \mathbb{X} \to \mathbb{X}$. This is colloquially referred to as *iterated function system* (IFS) (Barnsley, 2014). We can then define a map $F : \mathcal{H}(\mathbb{X}) \to \mathcal{H}(\mathbb{X})$ by

$$F(\mathbb{A}) = \bigcup_{k=1}^{K} f_k(\mathbb{A}) \quad \forall \mathbb{A} \in \mathcal{H}(\mathbb{X})$$

where $f_k(\mathbb{A})$ is intended as $f_k(\mathbb{A}) = \{f_k(a) : a \in \mathbb{A}\}$. An interpretation is the following: $F$ produces as output a composition, or *collage*, of transformations applied to a subset $\mathbb{A}$ of $\mathbb{X}$.

## 2.3 The Inverse Problem: Data to IFS

*Given data $\mathbb{S}$, can we find a map $F$ with $\mathbb{S}$ as its fixed point?* This can be achieved by identifying a collection of maps $f_k : \mathbb{X} \to \mathbb{X}$ such that the following conditions hold

$$i. \quad F : \mathcal{H}(\mathbb{X}) \to \mathcal{H}(\mathbb{X}); \mathbb{A} \mapsto \bigcup_{k=1}^{K} f_k(\mathbb{A}) \text{ is contractive;}$$

$$ii. \quad \mathbb{S} \text{ is } \underline{\text{the}} \text{ fixed point of } F, \ \mathbb{S} = F(\mathbb{S}) = \bigcup_{k=1}^{K} f_k(\mathbb{S});$$

Note that, $F$ is contractive w.r.t the Hausdorff metric with Lipsichitz constant $L < 1$ iff all the maps $f_k$ are contractive w.r.t $d$ with constant $\ell_k < 1$. In such a case it holds $L = \max_k\{\ell_k\}$ and $F$ admits a unique fixed point. A classical result provides an indirect method to find such $F$.

**Theorem 1** (Collage Theorem (CT) (Barnsley and Demko, 1985))**.** *Let $(\mathbb{X}, d)$ be a complete metric space and let $f : \mathbb{X} \to \mathbb{X}$ be a $\ell$-Lipschitz contractive map with fixed point $x^* \in \mathbb{X}$. Then,*

$$d(x, x^*) \leq \frac{1}{1 - \ell} d(x, f(x)) \tag{2.1}$$

By applying the CT directly to $F$ using the Hausdorff metric $d_{\mathcal{H}}$, we have $d_{\mathcal{H}}(\mathbb{S}, \mathbb{A}^*) \leq \frac{1}{1-L} d_{\mathcal{H}}\left(\mathbb{S}, \bigcup_{k=1}^{K} f_k(\mathbb{S})\right)$. This means that we can upper bound the distance $d_{\mathcal{H}}(\mathbb{S}, \mathbb{A}^*)$ between data $\mathbb{S}$ and attractor $\mathbb{A}^*$ of $F$ via $d(\mathbb{S}, F(\mathbb{S}))$, which requires a single application of $F$ and is thus cheaper to evaluate. Even if it is not possible to stitch together transformed copies $f_k(\mathbb{S})$ to perfectly reconstruct the data $\mathbb{S}$, i.e. $d_{\mathcal{H}}\left(\mathbb{S}, \bigcup_{k=1}^{K} f_k(\mathbb{S})\right) \neq 0 \quad (\Leftrightarrow \mathbb{S} \neq F(\mathbb{S}))$, a smaller IFS Lipschitz constant $L$ of the IFS implies a lower distance between the data $\mathbb{S}$ and the attractor $\mathbb{A}^*$ of $F$, given a mismatch $d_{\mathcal{H}}(\mathbb{S}, F(\mathbb{S}))$. This, in turn, implicitly promotes the use of "very contractive" maps $f_k$ (i.e. with low $\ell_k$). We refer to the procedure of searching for an $F$ that minimizes the r.h.s of the CT bound as the *fractal data encoding* problem.

**A learning perspective of fractal data encoding**  Fractal data encoding problem can be translated into finding a parametric representation $f_k(\,\cdot\,; w_k), w \in \mathbb{R}^{n_w}$ for functions $f_k(\cdot)$ (e.g. neural networks with parameters $w_k$) where $w = (w_1, \ldots, w_K) \in \mathbb{W}$ are optimized to minimize a Hausdorff metric loss function $d_{\mathcal{H}}(\mathbb{S}, F(\mathbb{S}; w))$ naturally induced by the CT,

$$\min_{w \in \mathbb{W}} \; d_{\mathcal{H}}(\mathbb{S}, \textstyle\bigcup_{k=1}^{K} f_k(\mathbb{S}; w_k)) \tag{2.2}$$

To decode the data encoded in parameters $w$ of $F$ encodes in one samples any initial condition $\mathbb{A}_0$, and iterates $\mathbb{A}_{t+1} = F(\mathbb{A}_t)$ until convergence to $\mathbb{A}^* \approx \mathbb{S}$.

**IFS, PIFS and Beyond**  *Partitioned iterated function systems* (PIFS) (Jacquin et al., 1992) are a generalization of IFSs that can capture localized self-similarity by allowing each domain of a contraction map $f_k$ to be a different subset $\mathbb{A}_k \subset \mathbb{S}$. his introduces a significant challenge in 2.2: the optimization problem need now determine optimal (as measured by $d_{\mathcal{H}}$) domains $\mathbb{A}_k$ for each $f_k$ by searching across all possible subsets of $\mathbb{S}$, yielding an exploding combinatorial problem.

**Solving for affine IFS**  As with traditional approximation problems, there is a tension in the objective of fractal data encoding between the "expressiveness" of the class of functions, and the tractability of the optimization problem. The solution $w$ of (2.2) is an equivalent representation for $\mathbb{S}$ (up to $d_{\mathcal{H}}(\mathbb{S}, \mathbb{A}^*)$).

Existing methods based on the idea of (Barnsley and Demko, 1985) resolve this tension by considering (a) compression as a task, such that $w$ should be encodeable in the least number of bits possible and (b) affine functions $f_k(x; w_k) = a_k x + b_k$. With these choices, a solution to (2.2) can be found in closed-form (Fisher, 2012), and the parametrization $w$ results compact enough to be a valid compression code - only two floats for each $f_k$ in $F$, i.e. $w_k = (a_k, b_k) \in \mathbb{R}^2$.

There are a number of limitations we aim to address:

- Fractal data encoding is only considered as an intermediate step towards compression. We develop a learning-based approach to the solution of (2.2) for tasks beyond compression.

- Solving (2.2) on a collection of data as per (Fisher, 2012) is computationally expensive. We directly solve a collection of fractal data encoding problems in parallel via hypernetworks (Ha et al., 2016), effectively amortizing the cost.

- As noted by (Welstead, 1999; Fisher, 2012), for an (affine) IFS to provide a satisfactory solution to (2.2), the self-similarity property has to be global across the set $\mathbb{S}$. That is, the entire set $\mathbb{S}$ is made up of smaller copies of itself, or a part of itself, property that is rather rare in natural data: indeed, most images are only self-similar to a degree. To alleviate these restrictions, we develop `Collage` operators, a generalization of IFSs which can be broadly categorized as a soft-*partitioned iterated function system* (PIFS) (Jacquin et al., 1992).

## 3  Neural `Collages`

Moving forward, we treat Neural `Collages` algebraically. This allows us to discuss in detail the properties of a Neural `Collage`, including algorithmic differences with other iterated function systems. To do so we consider, instead of generic sets, data that can be expressed as simple ordered sets: in other words, as vectors. Images will be our recurring example, with the understanding that the entire discussion can readily be adapted to other modalities e.g. sequences.

**From generic sets to vectors**  Following the PIFS treatment of Øien and Lepsøy (1995), we focus our analysis on `Collage` composed of *affine* maps, operating on the space of discrete images of a given resolution with a total number $m$ of pixels each taking values in $\mathbb{R}$. Pixels of different channels are treated without loss of generality as different elements. This allows us to collect all $m$ pixel values in an *ordered*[2] vector $z \in \mathbb{R}^m$.

A type of subsets on images involves the formation of square patches. Let us assume that each image is partitioned into (1) $K$ non-overlapping *range cells* and (2) $N$ possibly-overlapping *domain cells*.

---

[2]with a specific predefined criterion, e.g. row-major ordering.

A range cell $\mathsf{R}_k$ is then of size $n_{r,k} \times n_{r,k}$, such that $m = \sum_{k=1}^{K} n_{r,k}^2$, and domain cells $\mathsf{D}_n$ are of size $n_{d,n} \times n_{d,n}$. With such coordinatization, the (affine) fixed-point map $F$ reduces to a structured affine contraction on $\mathbb{R}^m$.

**Definition 1** (Neural Collage Operator). *Consider a $m$-pixel image represented by the ordered vector $z \in \mathbb{R}^m$. Then, a Collage Operator is defined as the parametric linear map:*

$$F(z; w) = \sum_{k,n} \gamma_{k,n} a_{k,n} T_k P_{k,n} S_n z + \sum_{k,n} \gamma_{k,n} b_{k,n} T_k \mathbb{1} \tag{3.1}$$

- $S_n \in \mathbb{R}^{n_{d,n}^2 \times m}$ **selects** *a domain cell* $\mathsf{D}_n$ *of* $n_{d,n} \times n_{d,n}$ *pixels.*

- $P_{k,n} \in \mathbb{R}^{n_{r,k}^2 \times n_{d,n}^2}$ *is a* **pooling** *operator that shrinks the domain cell* $\mathsf{D}_k$ *into the size of the corresponding range cell* $\mathsf{R}_k$, *i.e. from* $n_{d,n} \times n_{d,n}$ *to* $n_{r,k} \times n_{r,k}$ *pixels;*

- $T_k \in \mathbb{R}^{m \times n_{r,k}^2}$ **positions** *the pooled domain cell in the correct range cell location and zeroes out the rest;*

- $a_{k,n}, b_{k,n} \in \mathbb{R}$ **scales** *and* **translates** *the value in each pixel of the pooled domain cell, respectively.*

- $\gamma_{k,n} \in \mathbb{R}$; *convex* **combination** *of affine outputs produced from all $N$ domains.*

*The parameters $w$ is the collection of all $a_{k,n}, b_{k,n}$ and the mixing weights $\gamma_{k,n}$.*

Collage operators represent each range cell $\mathsf{R}_k$ as a convex combination of pooled and scaled versions of all domain cells $\mathsf{D}_n$ translated block-wise by $b_n$. On individual range cells comprising the output, a symbolic representation can be given as

$$\mathsf{R}_k = \sum_n \gamma_{k,n} a_{k,n} \mathsf{D}_n + \sum_n \gamma_{k,n} b_{k,n}. \tag{3.2}$$

In the generic set formulation, these maps correspond to functions $f_k$ of $F$. Differently from a PIFS, each $f_k$ does not act on a different subset (domain cell) to produce $\mathsf{R}_k$. Instead, all $f_k$ aggregate affine transformations – parametrized by $a_{k,n}, b_{k,n}$ – on domains via $\gamma_{k,n}$. However, each $f_k$ is equipped with different mixing weights and different affine maps. Hence, a Collage in this form can be seen as a soft-PIFS.

**Introducing auxiliary domains**   A Collage step maps mixtures of all domains to each range, and assembles the ranges into its output. As Neural Collages are often optimized on datasets, rather than single data points, we posit that improvements in the expressiveness can be readily achieved by mixing additional dataset-level information through *auxiliary domains* $\mathsf{U}_v$.

**Definition 2** (Collage operator with auxiliary domains).

$$\hat{\mathsf{R}}_k = \mathsf{R}_k + \sum_{v=1}^{V} \gamma_{k,v} a_{k,v} \mathsf{U}_v$$

*In coordinates this translates to*

$$\hat{F}(z, u; w) = F_w(z; w) + \sum_{k=1}^{K} \sum_{v=1}^{V} \gamma_{k,v} a_{k,v} T_k P_{k,v} S_v u$$

*where $F_w(z)$ is (3.1) and $\mathsf{R}_k$ is (3.2), and the parameters $w$ include the coefficients $\gamma_{k,v}, a_{k,v}$ of the auxiliary domains $\mathsf{U}_v$, as well as $\gamma_{k,n}, a_{k,n}, b_{k,n}$.*

We consider different variants of $\mathsf{U}_v$, including: deterministic transformations of domain cells $\mathsf{D}_n$ e.g. rotations as per (Jacquin et al., 1992), learned cells directly parametrized and optimized for an objective, similar to feature maps in (Jaegle et al., 2021), and $\mathsf{U}_v$ produced by a neural network encoder. Specifics are provided in Section 4. A schematic of a single step of Collage is given in Fig. 3.

## 3.1 The Forward Problem: Collage to Data

Given a parametrization $w$ for the `Collage` operator $F_w$ and an initial image $z_0$, the attractor $z^* : z^* = F_w(z^*)$ can be recovered by iterating the fixed-point map

$$z_{t+1} = F(z; w) = A(w)z_t + b(w)$$

assuming $F$ to be a contraction w.r.t. the standard Euclidean metric on $\mathbb{R}^m$. This can be ensured by an appropriate choice of the coefficients $a_{k,n}, \gamma_{k,n}$. In particular, if all the mixing weights are such that $\sum_{n=1}^N \gamma_{k,n} = 1$ and $|a_{k,n}| < 1$, then contractivity of the collage operator follows as in standard PIFS (see e.g. Fisher (2012)).

Note that the attractor of a `Collage` can be also computed in closed-form as $z^* = [\mathbb{I} - A(w)]^{-1}b(w)$. A similar discussion follows for a general `Collage` with auxiliary domains. Note that auxiliary domains $\mathsf{U}$ across iterations $t$ are to be chosen such that the sequence $\{\mathsf{U}_t\}_{t=0}^\infty$ converges e.g. constant functions.

Pseudocode for a single step of a `Collage` is shown below. Figure 2 provides a visualization of the convergence of a `Collage` to its fixed-point (after repeated application of the operator).

```python
def collage_operator(self, z, collage_weight, collage_bias):
    """Collage Operator (decoding). Performs the steps described in  Def. 3.1, Figure 2."""
    # Split the current iterate `z` into source patches according to the partitioning scheme.
    domains = img_to_patches(z)
    # Pool domains (pre augmentation) to range patch sizes.
    pooled_domains = pool(domains)
    # If needed, produce additional candidate source patches as augmentations of existing
    # domains, or concatenate auxiliary patches parametrized and optimized directly.
    if self.n_aug_transforms > 1:
        pooled_domains = self.generate_candidates(pooled_domains)
    pooled_domains = repeat(pooled_domains, 'b c d h w -> b c d r h w', r=self.num_ranges)
    # Apply the affine maps to source patches
    range_domains = einsum('bcdrhw, bcdr -> bcrhw', pooled_domains, collage_weight)
    range_domains = range_domains + collage_bias[..., None, None]
    # Reconstruct data by composing the output patches back together.
    z = patches_to_img(range_domains)
    return z
```

**Decoding at higher resolutions**   A `Collage` can be applied, without change, to images of different resolutions. Consider scaling the resolution by a factor $s > 1$, $(s \in \mathbb{N})$. Then the *magnified* image representation is made up of $s^2$ images $z^i \in \mathbb{R}^m$. The `Collage` operator can then be thought to act singularly on each $z^i$ obtaining a forward fixed-point iteration

$$z_{t+1}^i = \sum_{k,n} \gamma_{k,n}^i a_{k,n}^i T_k P_{k,n} S_n z_t^i + \sum_{k,n} \gamma_{k,n}^i b_{k,n}^i T_k \mathbb{1}.$$

When solving by unrolling the fixed-point iteration, the second term can be precomputed as it does not depend on $z_t$.

## 3.2 Amortized Solution of the Inverse Problem

Although the CT suggests a constructive procedure via (2.2) to find a valid fractal representation $w$, there are no guidelines in case other objectives are of interest. Further, the class of PIFS – without modifications – does not lend itself well to numerical optimization as it involves the combinatorial problem of matching domains to ranges. With their soft aggregation, Neural `Collages` can instead be used for task-based optimization. Given an input image $x \in \mathbb{R}^m$,

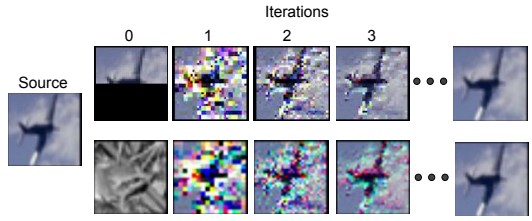

Figure 2: `Collage` forward problem: given $w$, different initial conditions decoded to the same $z^*$.

we can optimize the parameters $w$ (and pixel values $u$ of the auxiliary domains) of the Neural `Collage` operator to minimize an objective $J_\omega(x, z^*(w, u))$ by solving a nonlinear program $\min_{w,u} J_w(x, z^*(w, u))$ with $z^*(w, u)$ obtained by the fixed point iteration on $\hat{F}(z, u; w)$. Choosing $J_w$ to be a reconstruction objective yields a problem similar to fractal data encoding (2.2).

Suppose instead to be given an image dataset whose distribution $p$ is known only through i.i.d. samples $x \in \mathbb{R}^m$, $x \sim p(x)$. In this context, the fractal data encoding problem in standard form needs to be solved for each sample $x$. Instead, we introduce an *hypernetwork* (Ha et al., 2016) $\mathcal{E}$ with weights $\theta$, generating a fractal code $w$ given an input image $x$, i.e. parametrizing a map $x \mapsto w_\theta(x)$,

$$\forall x \sim p(x) \quad w_\theta(x) = \mathcal{E}(x; \theta)$$

The hypernetwork is trained to solve the following empirical risk minimization problem, effectively amortizing the cost over the full dataset:

$$
\begin{aligned}
\min_{\theta, u} \quad & \mathbb{E}_{x \sim p(x)}\left[J(x, z^*(\theta, u))\right] \\
\text{subject to} \quad & w_\theta(x) = \mathcal{E}(x; \theta) \\
& z^*(\theta, u) = \hat{F}(z^*(\theta, u), u; w_\theta(x)) \\
& u \in \mathbb{U}.
\end{aligned}
\tag{3.3}
$$

# 4  `Collages` **in Learning Tasks**

We showcase applications of Neural `Collages` as decoders for deep generative models, as neural compressors for images and as a method to generate fractal art. All variants of the model share a common structure, outlined in Figure 3. The code is available at `github.com/ermongroup/self-similarity-prior`.

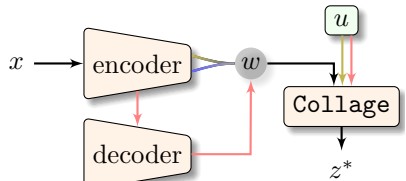

Figure 3: Schematic of a Neural `Collage`. In yellow, computation unique to the compressor variant; $x$ is deterministically encoded into `Collage` parameters $w$, then used to decode. Similarly, blue indicates computation done by the fractalizer, and red by the generative variant, where $w$ takes the role of a latent variable.

## 4.1  **Generative Neural** `Collages`

We investigate application of Neural `Collages` as deep generative models for distributions $p(x)$ of images. In this context, $w$ assume the role of latent variables. In particular, we consider a hierarchical *variational autoencoder* (VDVAE) (Kingma and Welling, 2013; Child, 2020) model based on Neural `Collages`.

**Magnifying samples via `Collage VAEs`**  VAE models seek to data $x$ into a latent representation $w$ such that the following lower bound (`ELBO`) on data log-likelihood be maximized

$$\mathbb{E}_{q_\phi}\underbrace{[\log p_\theta(x|w)]}_{-\text{distortion}} - \underbrace{D_{KL}[q_\phi(w|x)||p_\phi(w)]}_{rate} \leq \log p(x; \theta, \phi)$$

where approximate posterior $q_\phi(w|x)$, prior $p_\phi(w)$, and generator $p_\theta(x|w)$ are implemented as neural networks. A multiplicative hyperparameter $\beta$ is often introduce to control the relative weight between

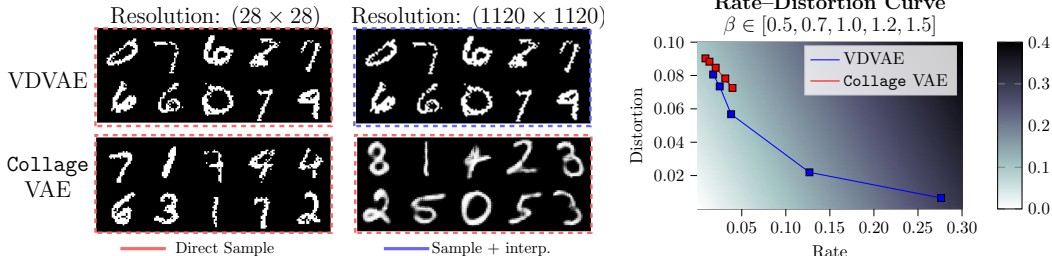

Figure 4: **[Left]** Samples obtained from VDVAEs and `Collage` VAEs on dynamically binarized MNIST. `Collage` VAEs can generate samples at resolutions unseen during training, adding detail that cannot be obtained by (1) sampling first (2) upscaling via bicubic interpolation, as done in the first row. **[Right]** Rate-distortion curve on dynamically binarized MNIST, with a curve obtained by sweeping $\beta$ in a range. `Collage` VAEs are less sensitive to the tuning of $\beta$, and pay a marginal price in Pareto suboptimality to gain the ability to sample at different resolutions.

rate and distortion (Higgins et al., 2016). In a `Collage` VAE, $q_\phi(w|x)$ and $p_\phi(w)$ are parameterized as per (Child, 2020), except the approximate posterior $q_\phi(w|x)$ need not produce large feature maps but scalar `Collage` codes $w$. Moreover, $p_\theta(x|w)$ is a `Collage` such that by leveraging the consideration of Section 3.1, samples can be decoded at any resolution.

**Results**   To investigate the properties of `Collage` VAEs, including quality of magnified samples, we compare VDVAEs and `Collage` VAEs on dynamically binarized MNIST. We report both test rate and distortion, following the analysis of (Alemi et al., 2018). This supports a more detailed evaluation of model behavior at different $\beta$ weights for the rate. We train several `Collage` VAEs and VDVAEs with $\beta$ ranging in $[0.5, 0.7, 1.0, 1.2, 1.5]$, and report the rate-distortion curve in Figure 4 (right). `Collage` VAEs are only marginally Pareto suboptimal, but are shown to be less sensitive to training under different $\beta$, which is a common strategy employed to stabilize VAE training ("KL warmup"). Furthermore, `Collage` VAEs can generate samples at resolution unseen during training, as showcased in Figure 4 (left). Direct samples at a magnification factor of $40\times$ reveal additional details over VDVAE samples that are magnified via bicubic interpolation. We note that as discussed in Section 5.1, the detail introduced by magnification is entirely dependent on the `Collage` class; future designs may be developed to introduce types of detail expected in a given dataset.

## 4.2   Neural `Collage` Compressors

Next, we apply Neural `Collage` to image compression. We store images as the parameters $w$ of an affine `Collage` produced by a neural encoder. After training, the encoder can be used to compress additional images in parallel with a single forward pass by producing the corresponding $w$. Figure 3 provides an overview of the computation done by a Neural `Collage` compressor. `Collage` compressors employ learned feature maps as auxiliary domains.

For compression, the main desideratum is visual fidelity: regular domains $\mathsf{D}_n$ allow Neural `Collages`

| Method | ↑ **PSNR** ╎ **bpp** ↓ | |
|---|---|---|
| Fractal (no aug.) | 31.06 ╎ 0.14 | 31.68 ╎ 0.36 |
| Fractal (augment.) | 30.51 ╎ 0.15 | 30.80 ╎ 0.39 |
| COIN | 25.74 ╎ 0.17 | 27.34 ╎ 0.33 |
| Neural `Collage` (**ours**) | 31.30 ╎ 0.13 | 32.12 ╎ 0.31 |
| block-DCT | 33.22 ╎ 0.13 | 34.65 ╎ 0.32 |

Table 1: Average *peak signal-to-noise ratio* (PSNR) at low ($\approx 0.15$) and medium ($\approx 0.30$) *bits-per-pixel* (bpp) budgets of baselines (baseline fractal, implicit, spectral) and Neural `Collages` compressors. Neural `Collages` introduces less visible artifacts than other self-similarity or implicit compression schemes, narrowing the gap with spectral compression methods such as JPEG.

to capture intra-image self-similarity, whereas auxiliary domains $\mathsf{U}$ optimized for image quality complement them by focusing on inter-image patterns.

**Compression of high-resolution aerial images**   We consider compressing images obtained from the DOTA large-scale aerial images dataset (Xia et al., 2018). From the DOTA training set, we produce

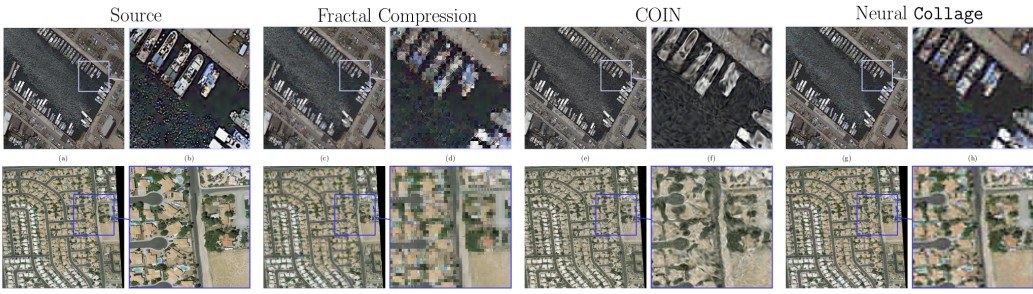

Figure 5: Visual comparison of decoded images from $1200 \times 1200$ crops of the DOTA dataset, obtained from different compression methods ($\approx 0.30$) bpp. Images decoded from Neural `Collage` codes exhibit less noticeable artifacts, with improved color preservation over COIN and more detail than fractal compression.

$80000$ random $40 \times 40$ crops as our training dataset. We optimize convolutional encoder parameters $\theta$ and auxiliary sources on the reconstruction objective $J(x, z^*(\theta, u)) = \sum_{i=1}^{m}(x_i - z_i^*(\theta, u))^2 + \|w\|_2^2$ The model is trained on the $40 \times 40$ crops and evaluated on 10 held-out $1200 \times 1200$ images. This is possible as an image of any resolution can be first broken up into blocks of appropriate size, in this case the training resolution, $40 \times 40$, passed through the encoder to obtain the corresponding parameters, then concatenated to construct a valid code for the entire image. This operation can be performed in parallel by treating each block as an element of a pseudo-batch. Neural `Collage` compressor can thus be used to compress images of any resolution at test time, without retraining.

**Results**   We compare *peak signal-to-noise ratio* (PSNR), in addition encoding and decoding measurements for a variety of compression baselines. We contextualize our results with comparisons to both non-neural as well as another implicit neural compressor. In particular, we evaluate the performance of a standard PIFS-based fractal compression as per (Jacquin, 1993), implemented to exploit parallelization on GPU, and COIN (Dupont et al., 2021). Fractal compression baselines and `Collage` both use non-adaptive tiling partitioning schemes. We evaluate two variants of fractal compression, one where domain cells are augmented via rotations and color flips (Welstead, 1999), and one without augmentations. We further compare with block-DCT, the spectral lossy compression backbone of most JPEG codecs. All compression methods are evaluated at low and medium bpps, with metrics provided in Table 1. Figure 5 provides a visual comparison of images obtained by decoding the lossy code of each method. Neural `Collages` show less noticeable artifacts and improved color retention.

Finally, Table 1 provides wall-clock time measurements of all methods during the respective per image encoding and decoding procedures. Results for COIN and Neural `Collage` measure encoding times (including the training procedure), and encoding times after training. Neural `Collages` are orders of magnitude faster than fractal compression and, at test-time, of COIN. Although spectral lossy compressors common in state-of-the-art codecs perform with best PSNR in medium and high bpp settings, `Collages` narrows the gap in terms of reconstruction quality as well as encoding speed.

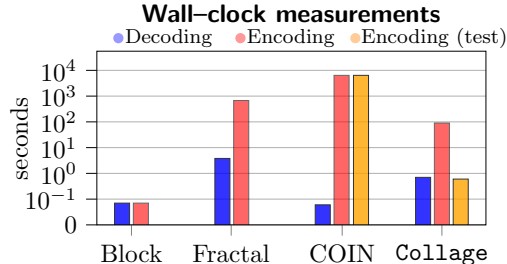

Figure 6: Wall-clock time (s) measurements of compression methods applied to the held-out set of DOTA. For neural compressors, we report encoding times including training, as well as encoding times on unseen images (after training). At test time, encoding for Neural `Collage` compressors is $100\times$ faster than fractal compression.

**Code length of a** `Collage`   The total *bits-per-pixel* (bpp) cost of the code $w$ corresponding to the parameters of a `Collage` depends on the coding scheme used to store each numeric entry. We exploit bounds on $a, b, \gamma$, enforced via `tanh` and softmax, as well as regularization to reduce the total cost. The $L_2$ regularization term on $w$ is introduced to shrink the range of values assumed by elements of $w$, ensuring that less bits can be used for storage. We do not use lossless coding schemes to store parameters of `Collages` and other baselines. Further details are provided in the Appendix.

### 4.3   Neural `Collages` for Fractal Art

When data is represented through the parameters $w$ of a `Collage`, it can be arbitrarily magnified by decoding at any resolution. The type of patterns revealed through magnification need not be corresponding to real detail missing from the image. In particular, the patterns found depend on how one generates domains, auxiliary domains and class of operator $F_w$. Similar phenomena have been observed in the literature of fractal compression (Mitra et al., 2000). As an example, consider Figure 7, where the fractal pattern of snowflakes within snowflakes does not correspond to reality. We call this type of globally self-referential magnification as fractalization of an image. Here, we use Neural `Collage` fractalizers to generate fractal art.

**Experimental details**   We solve the inverse problem of a `Collage`, namely image $x$ to $w$, via a convolutional architecture parametrized by $\theta$. The objective of the inverse problem (3.3) is a reconstruction objective $J(x, z^*(\theta, u)) = \sum_{i=1}^{m}(x_i - z_i^*(\theta, u))^2$ where $z^*$ is the fixed-point of the Neural `Collage` with parameters $w$ computed through the encoder $w = \mathcal{E}_\theta(x)$. We choose the single

domain D to be the entire image. Before applying $F_w$, we augment the domain via rotations of itself, utilizing those as auxiliary domains. Figure 1 and 7 show example fractalizations possible with Neural `Collages`, on greyscale and RGB images. The images can be magnified to any resolution (up to memory limits), revealing multiple fractal levels.

## 5 Related Work and Conclusion

**Implicit representations and models** Representation of data implicitly through functions is extensively used in simulation (Osher et al., 2004). (Sitzmann et al., 2020; Mildenhall et al., 2020; Dupont et al., 2021) parametrize the implicit functions via neural networks for use in downstream tasks such as compression. Implicit models, on the other hand, solve optimization problems within their forward pass (Poli et al., 2020). Neural `Collages` belong to both classes of methods: a fixed–point iteration with parameters defining data implicitly. In particular, Neural `Collages` can be framed as a compactly–parametrized operator variant of *deep equilibrium networks* (DEQs) (Bai et al., 2019), with parameters produced by a hypernetwork (Ha et al., 2016).

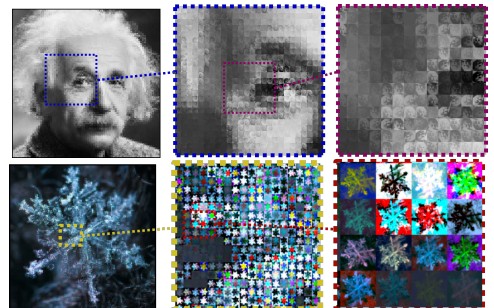

Figure 7: Fractalization of greyscale and RGB images via `Collage`. The images are compressed as the coefficients of a `Collage`, and decoded as its attractor. Note that the above is a showcase of *global* fractalization: no partitioning is performed, and the entire image itself is taken as the source.

**Fractal compression** The idea of representing images through *iterated function systems* (IFS) dates back to (Barnsley and Demko, 1985; Barnsley, 1986). Jacquin et al. (1992) introduces more flexible fractal compression schemes for images based on *partitioned iterated function systems* (PIFSs). Since then, alternative partitioning schemes i.e. adaptive quadtrees have been proposed. We refer to (Fisher, 2012) for an overview of the main variants. Additional references are provided in the Appendix.

**Conclusion** This work builds a framework for a learning–based approach to automated discovery of self–similarity. We introduce Neural `Collages`, implicit models designed to represent and manipulate data as the parameters of a structured fixed–point iteration, and showcase their application to compression and deep generative modeling of images. We envisage future use of Neural `Collage` with other data modalities with naturally occurring self–similarity, such as audio, sequences, or turbulent flows.

## Acknowledgments

This work is supported by NSF (1651565), AFOSR (FA95501910024), ARO (W911NF-21-1-0125), ONR, DOE, CZ Biohub, and Sloan Fellowship.

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
