# Self–Similarity Priors
## *Supplementary Material*

## A    Background and Extended Formulation

### A.1    Metric Spaces

**Lemma 1** (Useful results on bounded and closed sets)**.** *The following hold:*

 *i.* *Let $\mathbb{X} = \bigcup_{i=1}^{n} \mathbb{A}_i$ such that $\mathbb{A}_i$ is a **closed** subset of $\mathbb{R}^n$ for all $i = 1, \ldots, n$. Then, $\mathbb{X}$ is **closed**.*

 *ii.* *Let $\mathbb{X} = \bigcup_{i=1}^{n} \mathbb{A}_i$ such that $\mathbb{A}_i$ is a **bounded** subset of $\mathbb{R}^n$ for all $i = 1, \ldots, n$. Then, $\mathbb{X}$ is **bounded**.*

 *iii.* *Let $f : \mathbb{R}^n \to \mathbb{R}^n$ to be a continuous function and let $\mathbb{A}$ to be a closed, bounded subset of $\mathbb{R}^n$. Then $f(\mathbb{A}) = \{f(x) : x \in \mathbb{A}\}$ is also closed and bounded.*

**Definition 3** (Metric space)**.** *A metric space is a pair $(\mathbb{X}, d)$ where $\mathbb{X}$ is a set and $d : \mathbb{X} \times \mathbb{X} \to \mathbb{R}$ is a map such that for all $x, y, z \in \mathbb{X}$ the following conditions hold:*

      *i.* $d(x, y) \geq 0;$          *iii.* $d(x, y) = d(y, x);$

      *ii.* $d(x, y) = 0; \ \Leftrightarrow x = y$     *iv.* $d(x, y) \leq d(x, z) + d(z, y).$

*The function $d$ is called "metric". Note that a metric space is called **compact** if $\mathbb{X}$ is closed and bounded.*

**Definition 4** (Convergent sequence)**.** *Given a metric space $(\mathbb{X}, d)$, a sequence $\{x_t\}_{t=0}^{\infty}$ is said to converge to some $x^* \in \mathbb{X}$*

$$\forall \epsilon > 0 \ \exists t^* \ : \ \forall t > t^* \ d(x^*, x_t) < \epsilon.$$

**Definition 5** (Cauchy sequence)**.** *A sequence $\{x_t\}_{t=0}^{\infty}$ is $\mathbb{X}$ is a Cauchy sequence if*

$$\forall \epsilon > 0 \ \exists t^* \ : \ \forall s, t > t^* \ d(x_s, x_t) < \epsilon.$$

**Definition 6** (Complete metric space)**.** *A metric space $(\mathbb{X}, d)$ is complete if every Cauchy sequence in $\mathbb{X}$ is convergent in $\mathbb{X}$.*

**Definition 7** (Hausdorff space)**.** *Let $(\mathbb{X}, d)$ be a complete metric space, and define $\mathcal{H}(\mathbb{X})$ as the set of all compact subsets of $\mathbb{X}$:*

$$\mathcal{H}(\mathbb{X}) = \{\mathbb{A} \subset \mathbb{X} : \mathbb{A} \text{ is compact}\}.$$

**Definition 8** (Hausdorff metric)**.** *Let $(\mathbb{X}, d)$ be a metric space and let $\mathbb{Y}, \mathbb{Z} \subset \mathbb{X}$. The **Hausdorff metric** $d_{\mathcal{H}} : \mathcal{H}(\mathbb{X}) \times \mathcal{H}(\mathbb{X}) \to \mathbb{R}$ is then defined as*

$$d_{\mathcal{H}}(\mathbb{Y}, \mathbb{Z}) = \max \left\{ \sup_{y \in \mathbb{Y}} d(y, \mathbb{Z}), \sup_{z \in \mathbb{Z}} d(\mathbb{Y}, z), \right\}$$

*where $d(y, \mathbb{Z}) := \inf_{z \in \mathbb{Z}} d(y, z).$*

**Theorem 2** (Completeness of Hausdorff metric space (Barnsley and Hurd, 1993))**.** *Let $(\mathbb{X}, d)$ be a complete metric space. Then $(\mathcal{H}(\mathbb{X}), d_{\mathcal{H}})$ is a complete metric space.*

(Barnsley and Hurd, 1993) calls $(H(\mathbb{X}), d_{\mathcal{H}})$ *the space where fractals live*. It is the space where the mathematical foundations necessary to generate fractals via iterated function systems are developed.

## A.2 Contraction mappings

**Definition 9** (Lipschitz function). *Let $(\mathbb{X}, d)$ be a metric space. A map $f : \mathbb{X} \to \mathbb{X}$ is Lipschitz with constant $\ell$ if there exists $\ell > 0$ such that*

$$\forall x, y \in \mathbb{X} \quad d(f(x), f(y)) \le \ell d(x, y).$$

A Lipschitz function is then called *contractive* iff $\ell < 1$. Moreover, if $f : \mathbb{X} \to \mathbb{X}$ is Lipschitz, then $f$ is continuous.

> **Note:** A map $f$ is contractive if brings any elements of $\mathbb{X}$ *close together*. The Lipschitz constant $\ell$ measures (bounds) how closer the points are brought together by one application of $f$. Thus, intuitively, if we iterate a discrete dynamical system
>
> $$x_{t+1} = f(x_t)$$
>
> starting from any point of $x_0$ within $\mathbb{X}$, the sequence $\{x_t\}_{t=0}^{\infty}$ will converge to a **unique** fixed point $x^*$ such that $f(x^*) = x^*$.
> It is worth to notice that linear affine maps $f : \mathbb{R}^n \to \mathbb{R}^n$ (where we consider the standard Euclidian distance to form a complete metric space on $\mathbb{R}^n$)
>
> $$f(x) = Ax + b$$
>
> are contractive if and only if
>
> $$\|A\|_2 = \sup_{x \in \mathbb{R}^n} \frac{\|Ax\|_2}{\|x\|_2} < 1$$
>
> If this is the case any sequence $\{x_t\}_{t=0}^{\infty}$ would converge to a fixed point
>
> $$x^* = (I - A)^{-1} b.$$

The above intuitions can be formalized in the following classic result

**Theorem 3** (Banach fixed-point theorem). *Let $(\mathbb{X}, d)$ be a (non-empty) complete metric space and let $f : \mathbb{X} \to \mathbb{X}$ be a contractive map. Then $f$ admits a **unique** fixed point $x^* \in \mathbb{X}$ such that any sequence $\{x_t\}_{t=0}^{\infty}$ defined by the iteration*

$$x_{t+1} = f(x_t)$$

*converges to $x^*$ for any starting point $x_0 \in \mathbb{X}$, i.e.*

$$\exists! x^* \in \mathbb{X} \quad \lim_{t \to \infty} x_t = \lim_{t \to \infty} f(x_t) = x^* \quad \forall x_0 \in \mathbb{X}.$$

**Corollary 2** (Collage theorem). *Under the assumptions of Theorem 3, it holds*

$$\forall x \in \mathbb{X} \quad d(x, x^*) \le \frac{1}{1 - \ell} d(x, f(x)).$$

## A.3 Iterated Function Systems

With the aim of deriving fractal compression algorithms it is necessary to define functions on the Hausdorff metric space $(\mathcal{H}(\mathbb{X}), d_{\mathcal{H}})$. In particular, let $\{f_1, f_2, \ldots, f_K\}$ be a collection of maps on $\mathbb{X}$, $f_k : \mathbb{X} \to \mathbb{X}$. Then, we can define maps $F : \mathcal{H}(\mathbb{X}) \to \mathcal{H}(\mathbb{X})$ by

$$F(\mathbb{A}) = \bigcup_{k=1}^{K} f_k(\mathbb{A}) \quad \forall \mathbb{A} \in \mathcal{H}(\mathbb{X})$$

where $f_k(\mathbb{A})$ is intended as $f_k(\mathbb{A}) = \{f_k(a) : a \in \mathbb{A}\}$. Moreover, note that $\mathbb{A} \in \mathcal{H}(\mathbb{X})$ is also a compact subset of $\mathbb{X}$. The following results shows that if all $f_k$ are contractive, then also $F$ is.

**Theorem 4** (Contractivity of maps on the Hausdorff metric space). *If for all $k = 1, \ldots, K$, the maps $f_k : \mathbb{X} \to \mathbb{X}$ are contractive with Lipschitz constant $\ell_k < 1$, then $F : \mathcal{H}(\mathbb{X}) \to \mathcal{H}(\mathbb{X}); \mathbb{A} \mapsto \bigcup_{k=1}^{K} f_k(\mathbb{A})$ is contractive in the Hausdorff metric with Lipschitz constant $L = \max_k \{\ell_k\}_k$.*

**Definition 10** (Iterated function system (IFS)). *An iterated function system is a collection $\{f_1, \ldots, f_K\}$ of contractive maps $f_k : \mathbb{X} \to \mathbb{X}$, represented by $F : \mathcal{H}(\mathbb{X}) \to \mathcal{H}(\mathbb{X})$.*

Thanks to its contractivess (as established by Theorem 4), $F$ define a unique fixed point (*attractor*) by the *Banach fixed-point theorem* (Theorem 3), i.e. the discrete iteration defined by

$$\mathbb{A}_{t+1} = F(\mathbb{A}_t) = \bigcup_{k=1}^{K} f_k(\mathbb{A}_t)$$

converges to $\mathbb{A}^* \in \mathcal{H}(\mathbb{X})$ for $t \to \infty$. Since the attractor $\mathbb{A}^*$ is unique, it is completely defined by the map $F$. The *data encoding problem* can be then formulated as follows.

---

**Problem:** *Fractal Data Encoding* (Section 2.2)

If we are given some set $\mathbb{S} \in \mathcal{H}(\mathbb{X})$ (our *data*), can we find map $F$ whose attractor is $\mathbb{S}$?

In other words, given data $\mathbb{S} \in \mathcal{H}(\mathbb{X})$, find the collection of maps $f_k : \mathbb{X} \to \mathbb{S}$ such that the following conditions hold

$$i. \quad F : \mathcal{H}(\mathbb{X}) \to \mathcal{H}(\mathbb{X}); \mathbb{A} \mapsto \bigcup_{k=1}^{K} f_k(\mathbb{A}) \text{ is contractive;}$$

$$ii. \quad \mathbb{S} \text{ is } \underline{\text{the}} \text{ fixed point of } F, \ \mathbb{S} = F(\mathbb{S}) = \bigcup_{k=1}^{K} f_k(\mathbb{S});$$

---

**Properties of data encoding**   Note that condition $ii.$ suggests that, in order for the problem to admit a solution, data should be made up of transformed copies of itself. Specifically, we are assuming that it is possible to take the data $\mathbb{S}$, copy it $K$-times, apply to the copies some contractive transformations and finally stitch them together to reconstruct the initial data $\mathbb{S}$. The uniqueness of an attractor induced by the contractivity of $F$ is fundamental to practically solve the fractal encoding problem because if we can find an $F$ such that $\mathbb{S} = F(\mathbb{S})$, then we will be sure that $F$ is the unique solution of the encoding problem.

**On the Collage Representation**   By applying the *Collage Theorem* (Corollary 2) to $F$ using the Hausdorff metric $d_{\mathcal{H}}$ we have

$$d_{\mathcal{H}}(\mathbb{S}, \mathbb{A}^*) \leq \frac{1}{1 - L} d_{\mathcal{H}}(\mathbb{S}, F(\mathbb{S}))$$

$$\Leftrightarrow \ d_{\mathcal{H}}(\mathbb{S}, \mathbb{A}^*) \leq \frac{1}{1 - \max_k\{\ell_k\}} d_{\mathcal{H}}\left(\mathbb{S}, \bigcup_{k=1}^{K} f_k(\mathbb{S})\right)$$

This means that if we can't stitch the transformed copies $f_k(\mathbb{S})$ together to perfectly reconstruct the data $\mathbb{S}$, i.e.

$$d_{\mathcal{H}}\left(\mathbb{S}, \bigcup_{k=1}^{K} f_k(\mathbb{S})\right) \neq 0 \quad (\Leftrightarrow \mathbb{S} \neq F(\mathbb{S})),$$

then the lower the Lipschitz constant $L$ of the IFS is, the lower the distance between the data $\mathbb{S}$ and the attractor $\mathbb{A}^*$ of $\mathbb{S}$ will be given a mismatch $d_{\mathcal{H}}(\mathbb{S}, F(\mathbb{S}))$. As mentioned in the main text, this implicitly promotes the use of "very contractive" maps $f_k$ (i.e. with low $\ell_k$).

**A learning perspective to the *fractal data encoding***   In the language of machine learning practice, the *fractal data encoding* problem can be translated into finding a parametric representation $f_k(\,\cdot\,; w_k)$, $w \in \mathbb{R}^{n_w}$ for the functions $f_k(\cdot)$ (e.g. Neural Networks with parameters $w_k$) where the parameters $w = (w_1, \ldots, w_K) \in \mathbb{W}$ are trained to minimize the Hausdorff metric loss function $d_{\mathcal{H}}(\mathbb{S}, F(\mathbb{S}; w))$

naturally induced by the *Collage Theorem*, i.e.

$$\min_{w} \quad d_{\mathcal{H}}(\mathbb{S}, F(\mathbb{S}; w))$$

$$\text{subject to} \quad F(\mathbb{S}; w) = \bigcup_{k=1}^{K} f_k(\mathbb{S}; w_k)$$

$$w \in \mathbb{W}$$

Once optimal $f_k( \cdot ; w_k)$ are computed, it is easy to find the data that $F$ encodes (i.e. the *decoding* process): after sampling any initial condition $\mathbb{A}_0$, the encoded data can be obtained by iterating

$$\mathbb{A}_{t+1} = F(\mathbb{A}_t),$$

until convergence to $\mathbb{A}^* \approx \mathbb{S}$.

### A.4 Partitioned Iterated Function Systems

Existance of solutions of the fractal encoding (inverse) problem requires data to be perfectly representable by an IFS. Conversely, we can define *self-similar* data if it is the attractor of an IFS.

**Definition 11** (Self-similar sets). *A set $\mathbb{S}$ is called self-similar if and only if there exists a contractive map $F : \mathcal{H}(\mathbb{X}) \to \mathcal{H}(\mathbb{X})$ whose attractor is $\mathbb{S}$, i.e. $\mathbb{S} = F(\mathbb{S})$.*

While verifying the self-similarity of a specific data point is undoubtedly a NP-hard problem, natural data (e.g. in an image dataset) is unlikely to satisfy this strict property. The challenge is that the self-similarity property has to be global across the set $\mathbb{S}$. That is, the entire set $\mathbb{S}$ has to be made up of smaller copies of itself, or parts of itself. If one zooms in on it, it would display the same level of detail, regardless of the resolution scale (Welstead, 1999).

For this reason, it is necessary to extend the fractal encoding to more general, non globally self-similar sets. This can be achieved by introducing the technology of *partitioned function systems* (PFS) where the domains of the contraction maps $f_k$ are restricted.

**Definition 12** (Partitioned Function System). *Let $(\mathbb{X}, d)$ be a complete metric space and let $\mathbb{D}_k \subset \mathbb{X}$ for $k = 1, \ldots, K$. A partitioned function system is a collection of contraction maps $f_k : \mathbb{D}_k \to \mathbb{X}$.*

Note that, according to Fisher (2012), it is not possible to extend Theorem 3 to PFSs in the general case to effectively ensure existance and uniqueness of fixed points. Intuitively this is due to the fact that the domains of $f_k$ are restricted and the convergence of the decoding dynamics (i.e. the fixed-point iteration)

$$\mathbb{A}_{t+1} = \bigcup_{k=1}^{K} f_k(\mathbb{A}_t)$$

becomes dependent on the choice of the initialization $\mathbb{A}_0$. In fact, even though if choose $\mathbb{A}_0 \subset \bigcap_{k=1}^{K} \mathbb{D}_k$, after one step we may end up with an empty set. Note that this is generally not a problem in practice when applying PFSs to or `Collage` working with common type of data such as images or audio signals.

### A.5 Functional Representation of Data

In order to derive an implementation-oriented formulation of data encoding with partitioned functions systems in the general case, it can be convenient to rely on a *functional* description of data (see e.g. (Welstead, 1999; Fisher, 2012)). In example, images (of infinite resolution) can be represented as functions from the unit square to $\mathbb{R}$. Time series can also be thought as real continuous functions over a compact time domain.

Specifically we restrict our analysis to the space $\mathcal{F} = \{\phi : \text{dom}(\phi) \to \mathbb{R}\}$ of *data* defined the *graphs* $(z, \phi(z)), z \in \text{dom}(\phi)$ of (measurable) functions over the compact domain $\text{dom}(\phi)$ and values in $\mathbb{R}$. $\text{dom}(\phi)$ is assumed to be a compact subset of $\mathbb{R}^n$.

**Partitioned fractal encoding *a la* Welstead (1999)**    We choose $\mathcal{F} = L^p(\mathcal{X}; \mathbb{R})$ with $\mathcal{X}$ a compact subset of $\mathbb{R}^n$ and we equip it with a metric $d_\mathcal{F}$ induced by the Lebesgue measure

$$d_\mathcal{F}(\phi, \psi) = \left( \int_\mathcal{X} |\phi(x) - \psi(x)|^p \mathrm{d}x \right)^{1/p}.$$

Then $(\mathcal{F}, d_\mathcal{F})$ is a complete metric space and the *Banach fixed-point* (Theorem 3) holds. Then, we specialize the partitioned function system on $(\mathcal{F}, d_\mathcal{F})$ as comprised of the following collections of $K$ elements:

 a. *sub-domains* $\mathsf{D}_k \subset \mathcal{X}$;

 b. *invertible contractive maps* $v_k : \mathsf{D}_k \to \mathsf{R}_k \subset \mathcal{X}$;

 c. *maps* $f_k : \mathcal{F} \to \mathcal{F}$ *defined as*

$$f_k(\phi)(x) = c_k \phi(v_k^{-1}(x)) + d_k \quad \forall \phi \in \mathcal{F}, x \in \mathcal{X}.$$

Note that we can define subsets $\mathsf{R}_k$ as the range of $v_k$ operating on $\mathsf{D}_k$, i.e. $\mathsf{R}_k = v_k(\mathsf{D}_k)$. The constants $c_k, d_k$ realize an affine trasformation on $\phi$ by expanding/contracting and shifting the range of $\phi$. The contractive maps $v_k$ are the "spatial part" of the PFS and map the domains $\mathsf{D}_k$ to their respective ranges $\mathsf{R}_k$. $v_k$ are often chosen to be affine maps

$$v_k(x) = A_k x + b_k, \quad A_k \in \mathbb{R}^{n \times n}, b_k \in \mathbb{R}^n.$$

Note that it is possible to choose $A_k$ and $c_k$ so that $f_k$ is contractive. In particular, it is sufficient to require $|c_k||\det A_k|^{1/p} < 1$

**Definition 13** (Tiling partition). *A collection of ranges $\mathsf{R}_k \subset \mathcal{X}$ is said to tile $\mathcal{X}$ iff $\mathcal{X} = \bigcup_{k=1}^{K} \mathsf{R}_k$ and $\forall i \neq j \; \mathsf{R}_i \cap \mathsf{R}_j = \emptyset$.*

If the ranges $\mathsf{R}_k$ *tile* $\mathcal{X}$, we can define the operator $F : \mathcal{F} \to \mathcal{F}$ by

$$F(\phi)(x) = f_k(\phi)(x) \text{ for } x \in \mathsf{R}_k,$$

i.e.

$$F(\phi)(\mathcal{X}) = \bigcup_{k=1}^{K} f_k(\phi)(\mathsf{R}_k) = \bigcup_{k=1}^{K} c_k \phi(\mathsf{R}_k) + d_k = \bigcup_{k=1}^{K} c_k \phi(v_k^{-1}(\mathsf{D}_k)) + d_k$$

Since the ranges $\mathsf{R}_k$ tile $\mathcal{X}$, $F$ is defined for all $x \in \mathcal{X}$, so $F(\phi)$ is a function of the same class of $\phi$.

If $\phi$ is an image on the unit square tiled by the ranges $\mathsf{R}_k$, then $F(\phi)$ will also be an image on the unit square.

Assuming all the maps $f_k$ to be contractions on $\mathcal{F}$, $F$ satisfies the Banach fixed point theorem and has unique fixed point $\phi^* \in \mathcal{F}$ such that

$$\phi^* = F(\phi^*).$$

**Collage and PIFS for digital images**    Similarly to the main text, we can define PIFS by restricting our analysis to *affine* maps, operating on the space of discrete images of a given resolution with a total number $m$ of pixels. Note that pixels across channels can be treated effectively as different elements.

We assume the value of each pixel to range in $\mathbb{R}$ and to collect all the pixel values in an *ordered*[3] in a vector $z \in \mathbb{R}^m$. Then, a *partitioned function system* (Jacquin, 1993; Welstead, 1999; Fisher, 2012) can be represented as the *structured* map

**Definition 14** (Discrete PIFS). *Consider a $m$-pixel image represented by the ordered vector $z \in \mathbb{R}^m$. Then, a Discrete PIFS is defined as the parametric linear map $F : \mathbb{R}^m \to \mathbb{R}^m$:*

$$F(z; w) = \sum_{k=1}^{K} a_k T_k P_k S_k z + \sum_{k=1}^{k} b_k T_k \mathbb{1}, \quad w = (a_1, \dots, a_K, b_1, \dots b_K) \tag{A.1}$$

*where $T_k, P_k, S_k$ are defined similarly to Definition 1.*

---

[3] with a specific predefined criterion.

**Algorithm 1** Fractal Compression with PIFS Jacquin et al. (1992)
● Encoding ● Decoding

---

**Input:** Image $\mathsf{I}$, parametrized PIFS $F(\,\cdot\,; w)$.
Partition $\mathsf{I}$ into two collections of $N$ domains $\mathsf{D}_n$ and $K$ ranges $\mathsf{R}_k$.
**for** $k$ **from** $1$ **to** $K$ **do**
$\quad \mathsf{D}_n^*, w_k^* = \arg \min_{\mathsf{D}_n, w_k} d(f_k(\mathsf{D}_n; w_k), \mathsf{R}_k)$                    *matching*
**end for**
Store `code` $:= \{w_k^*, (n, k)\}_{k=1}^K$                    *fractal code and domain index*
Initialize $\tilde{\mathsf{I}}_0$ as any image with same size as $\mathsf{I}$
**repeat**

$$\tilde{I}_{t+1} = \bigcup_{k=1}^K f_k(\mathsf{D}_{n,t}^*; w_k^*)$$

**until** convergence
**return** $\tilde{I}^*$                    *decoded data*

---

Figure 8: Algorithmic summary of fractal compression with PIFS.

The output of the collage operator for $\mathsf{R}_k$ is thus a pooled and scaled version of $\mathsf{D}_k$ translated block-wise by $b_k$. A symbolic formulation of the collage operator can be also given by

$$\mathsf{R}_k = f_k(\mathsf{D}_k; w_k), \quad w_k = (a_k, b_k)$$

Since the collection of range cells $\mathsf{R}_k$ tiles the whole image $\mathsf{I}$, we can write (*a la* IFS)

$$F(\mathsf{I}; w) = \bigcup_{k=1}^K f_k(\mathsf{D}_k; w_k)$$

**Remark 1** (Extensive search). *Note that, in the classic setting of Jacquin (1993); Welstead (1999), for each range cell $\mathsf{R}_k$, the corresponding domain cell $\mathsf{D}_k$ has to be found by extensive search through the set of all possible pooled domain cells.*

We provide a compact algorithmic summary of the core steps in fractal compression as per Jacquin et al. (1992); Jacquin (1993) in Figure 8. Other variants of fractal compressions have historically been attempted, including ones with adaptive partitions and different algorithms to solve the combinatorial search during encoding.

# B  Additional Details

**Code-length flexibility: `Collage` and PIFS**  A `Collage` operator is a generalization of PIFS operators of fractal compression algorithms (Jacquin et al., 1992; Jacquin, 1993; Fisher, 2012). In particular, the `Collage` introduces additional flexbility in the choice of *code length* i.e. how many bits to allocate to the compression code. For example, given non-adaptive (square) tiling domain and range cells[4], the *bits-per-dimension* (bpp) cost of PIFS-based fractal compression of Jacquin et al. (1992) is $\frac{\texttt{cost}_{w_k}}{n_{r,k}^2}$, where $\texttt{cost}_{w_k}$ is the cost of saving the parameters of single element in the $f_k$ of $F_w$.

Here, other than seeking further compression and reduction of $\texttt{cost}_{w_k}$ by modifying the class of operator, the only degree-of-freedom is to reduce or increase the dimensions of tiling partitions. Note that modifying the partition scheme has not only effect on the bpp cost but also on the type of self-similarity that can be captured. Instead, `Collage` operators offer an additional design axis; indeed, the bpp budget – given a fixed partition – can be modified by increasing or decreasing the number of auxiliary domains introduced as learnable feature maps. This number is independent on the number of original domains, whereas the number of additional domains generated as affine augmentations of fractal compression (see Welstead (1999); Fisher (2012) for details) is not.

**On adaptive partitions**  Elaborated partitioning schemes have been developed for PIFS-based fractal compression methods (Fisher, 2012). While the analysis and empirical comparisons of this work have been centered around the operators, rather than partition schemes, we remark that `Collage` are compatible with alternative and potentially adaptive schemes. Much like for PIFS, this is a likely direction for further improvement of Neural `Collage`.

## B.1  `Collage` Schematic

**Collage Operator**

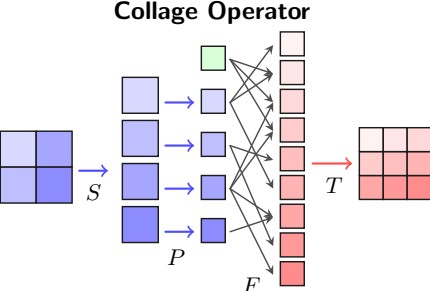

Figure 9: Conceptual schematic of a `Collage`. In blue, domain cells $S_n$; in red, range cells $R_k$ of ranges. Green highlights auxiliary domains. A step can be broken down into (1) $S$ partitions into domains (2) $P$ reduces dimensions to ensure dimensions match (3) $F$ produces all range cells, each following (3.2) (4) $T$ rearranges the output.

## B.2  Extended related work

**Attention operators and patches**  There exist superficial similarities between `Collages` and attention operators (Vaswani et al., 2017). In particular, recent variants of vision transformers (Dosovitskiy et al., 2020) where attention acts on square patches, can be seen as a single step of a `Collage`, where source and target partition match and aggregation weights are found via similarity scores. `Collages` differ from attention in that they are structured fixed-point iterations, are built to accommodate non-overlapping partitions, are resolution-invariant, and have a compact parametrization that can be used as a compression code. It remains to be seen whether investigating attention operators through the lenses of `Collages` can yield improvements in theoretical understanding or performance.

**Fractal compression**  Sun et al. (2001) parametrize elements of the iterative map with small neural networks. The proposed method still requires training on each image, with marginal improvements

---

[4]Although different choices are possible, "tiling" partitions are most convenient; when applying $F$, a domain partition D can be identified via single $\lceil \log_2 N \rceil$-bit integer address.

over standard variants. (Guido et al., 2006) provide a preliminary exploration of fractal coding for audio. Despite the extensive body of work, fractal methods for image compression are rarely used in place of other codecs due to slow encoding. As discussed in the main text sections, Neural `Collages` address this limitation via neural network amortization. We note that Neural `Collage` remain compatible with adaptive partitioning schemes, which provides a likely avenue of further improvement. We highlight a line of work on different probabilistic models of self-similarity (Zha et al., 2020) for tasks such as image restoration.

# C   Additional Experiment Details

**Hardware and software**   The experiments have been performed on a workstation with 2 NVIDIA GEFORCE RTX 3090 GPUs. We use JAX[5] for model implementation and distributed training.

## C.1   Neural `Collages` for Fractal Art

We construct the encoder $\mathcal{E}$ for $w$ by stacking 4 blocks composed of interleaved depthwise and pointwise convolutions. We train for 5000 iterations on each image displayed in Figure 7 with AdamW (Loshchilov and Hutter, 2017). We produce U by augmenting at each step of the `Collage` with rotations of $[90, 180, 270]$ degrees and flips, produced by multiplying all pixels values of a domain cell by $-1$. We do not use any additional learned auxiliary domain, so that the patterns can be kept globally fractal.

**Neural `Collage` Texturizers**   We report additional results in D.3, where a Neural `Collage` is used to texturize images by optimizing transformations of a fixed U, provided as external "texture source". To promote utilization of texture sources we introduce a coefficient to weigh U relative to D in the `Collage` iteration. We note that in this case the Neural `Collage` is not leveraging any self-similarity; rather, this should be intended as a display of the capability of a Neural `Collage` to aggregate both external as well as self-referential information to achieve a given task.

## C.2   Neural `Collages` for Generation

We design the architecture of a `Collage` following (Child, 2020). Table 3 describes the model structure. We introduce 30 learned auxiliary domains, parametrized to be pixel patches of same size of range cells. As the tiling partition, we choose a single domain cell of size $28 \times 28$ and size 4 range cells of $14 \times 14$. All models use a Bernoulli likelihood. Note that in this case, the fixed-point of the generator $p(x|w)$, chosen as a `Collage`, is given by the collection of all Bernoulli parameters, one for each pixel. This is thus an example of a `Collage` that is does not decode pixel-values of an image as its fixed-point, but rather parameters of their distributions. We optimize the `ELBO` by sweeping the KL weight $\beta$ as discussed in Figure 4 for 2000 epochs. Additional training details are provided in 4.

## C.3   Neural `Collages` for Compression

We construct the encoder $\mathcal{E}$ for $w$ by stacking 4 blocks composed of interleaved depthwise and pointwise convolutions. We train for 5 epochs with AdamW Loshchilov and Hutter (2017) on a dataset of 8000 crops of size $40 \times 40$ obtained from the DOTA Xia et al. (2018) aereal image training dataset. The dataset is generated (statically) randomly by applying a random rotation, followed by a random crop. We produce the 10 held-out images of size $1200 \times 1200$ with a similar procedure, applied to the test dataset. We note that DOTA images are all of different resolutions, motivating the above procedure. Speedup results are provided in Figure 12.

As baselines, we use the official COIN Dupont et al. (2021) implementation. We develop a GPU-parallel version of fractal compression with PIFS Jacquin et al. (1992); Jacquin (1993); Welstead (1999); Fisher (2012) as a baseline. We use the same partition strategy as for Neural `Collages`, namely tiling into domain and range cells. Our evaluation includes a fractal compression variant which incorporate U by augmenting domains, D at each step, with rotations of $[90, 180, 270]$ degrees and flips, produced by multiplying all pixels values of a domain cell by $-1$. The matching problem of domains to ranges is solved via least-squares as per Welstead (1999). We parallelize the least-square solving across domains.

## C.4   Computation of bits-per-pixel

We report the per-image *bits-per-pixel* (bpp) cost of compression baselines and Neural `Collage` compressors.

---

[5]https://github.com/google/jax

| | $K$ | $N$ | $V$ | $\epsilon$ | $\text{bpp}_u$ | $\text{bpp}_a$ | $\text{bpp}_b$ | total |
|---|---|---|---|---|---|---|---|---|
| low-bpp | 4 | 1 | 3 | 3 | $9 \cdot 10^{-4}$ | $6.6 \cdot 10^{-3}$ | $6.3 \cdot 10^{-3}$ | 0.134 |
| medium-bpp | 4 | 1 | 10 | 4 | $8.9 \cdot 10^{-4}$ | $6.5 \cdot 10^{-3}$ | $5.9 \cdot 10^{-3}$ | 0.319 |

Table 2: Example *bits-per-pixel* (bpp) code length computation for Neural `Collage`. To determine $\text{bpp}_a$ and $\text{bpp}_b$ we consider respective maximum values and represent their quantized integer range as discussed in C.4. The cost of auxiliary inputs is amortized on the 10 held-out $1200 \times 1200$ crops of DOTA, as decoding uses the same learned auxiliary domains for all images. We consider the auxiliary cost $\text{bpp}_u$ as part of the fractal code for a worst-case comparison, noting that reutilization of the same compressor eventually amortizes the cost to 0. For $M$ images, $\lim_{m \to \infty} \text{bpp}_u = 0$.

**Neural `Collage`** Computation of the bpp of fractal codes generated by a Neural `Collage` compressor requires the following considerations. First, mixing weights $\gamma_{k,n}$ are premultiplied to both $a_{k,n}$ and $b_{k,n}$. The same holds for mixing weights of auxiliary domains. We store a single $b_k$ for each $\mathsf{R}_k$, noting that the corresponding term can be precomputed as $b_k = \sum_{n=1}^{N} \gamma_{k,n} b_{k,n}$. Further, we exploit the a priori knowledge that $a, b \in [-1, 1]$, enforced via `tanh`, in combination with significant digit clipping. Quantizing by clipping to a threshold $\epsilon$ of significant digits, in combination with the bounds enforced by `tanh`, allows bit-packing each into less than $\lceil \log_2 10^\epsilon \rceil + 2$. One of the 2 additional bits is for sign information. This can be verified by noticing that by quantizing the range of values, after multiplying by $10^\epsilon$, is contained by the integer range $[-10^\epsilon, 10^\epsilon - 1]$. In practice, the number of bits is less than $\lceil \log_2 10^\epsilon \rceil + 2$ since not all values in entire integer interval defined by $\epsilon$-quantization are utilized by $a_{n,k}$ and $b_k$ for a given image. In particular, we consider maximum absolute values of the quantized values, and restrict the interval accordingly.

Neural `Collage` do not require storing of domain cell addresses, since each map of the `Collage` corresponding to a range cell (see 3.2) always transforms all domains. The specification of patch-sizes, especially when they are the same across all domains, and across all ranges, as well as type of pooling operators can be considered part of the codec, adding a negligible amount of bits. Further considerations are necessary in one wishes to employ more elaborate partition schemes (Fisher, 2012).

The overall cost is given by

$$\text{bpp} = K(N + V)\text{bpp}_a + K\text{bpp}_b + V\text{bpp}_u \tag{C.1}$$

where $N$ is the number of domains, $V$ is the number of learned auxiliary cells, and $K$ the number of ranges. We indicate with $\text{bpp}_u$ the cost of saving auxiliary learned patches. This cost is amortized across each image of the held-out set, as the patches are the same for a given Neural `Collage`. For $V$ auxiliary patches of size $h \times w \times c$, the (non-amortized) bit cost is $32 \cdot V \cdot h \cdot w \cdot c$ bits. We provide some example calculations in 2.

**COIN** We use the official implementation of (Dupont et al., 2021), where the bpp is computed by serializing the weights of the network into a bytes.

**Fractal compression baseline** We quantize fractal compression affine maps $f_k$ into half-floats, 16 bits for each $a_k$ and $b_k$. As the address of the domain cell associated to a given range $\mathsf{R}_k$, we store the index with cost $\lceil \log_2 (N + V) \rceil$-bit, where $N$ is the number of source domains $\mathsf{D}$ and $V$ the number of auxiliary domains $\mathsf{U}$.

**block-DCT** We apply a forward, two-dimensional *discrete cosine transform* (DCT) to patches of sizes 12 (high bpp) and 16 (medium bpp) and filter all but the lowest coefficient. The total cost is thus $32 \cdot n_{\text{patches}}$. The image is decoded by applying an inverse DCT.

| | ARCHITECTURE | | | LEARNING PARAMETERS | | |
|---|---|---|---|---|---|---|
| Model | ENCODER | DECODER | CHANNELS | $\beta$ | DECODER LATENT | NUM. AUXILIARY |
| Collage VAE | "28x1,28d4,7x1,7d7,1x1" | "1x4" | 128 | [0.5, 0.7, 1.0, 1.2, 1.5] | 64 | [30] |
| VDVAE | "28x1,28d4,7x1,7d7,1x1" | "1x1,7m1,7x2,28m7" | 64 | [0.5, 0.7, 1.0, 1.2, 1.5] | 16 | – |

Table 3: Autoencoder model setup for VDVAE and VDCVAE (ours). Both were trained on BMNIST with Bernoulli likelihood. The encoder and decoder architectures strings can be interpreted as follows: "28x1" indicates 1 block of 28 residual layers and "28m7" is the mixing a 7 by 7 activation into an upsampled 28 by 28 embedding. A block consists of standard autoencoder parameterization with a learned prior, posterior and latent projection output layer. The channel dimensions apply to both the widths of encoder and decoder blocks with the collage decoder variants having learned separate maps per channel. $\beta$ describes the KL-coefficient weighting in the ELBO objective.

| | TRAINING | | | | OPTIMIZER | | |
|---|---|---|---|---|---|---|---|
| Model | BATCH SIZE | EPOCHS | EMA | PER STEP (SECS) | LEARNING RATE | WEIGHT DECAY | OPTIMIZER |
| Collage VAE (ours) | 32 | 2000 | 0. | 1.7 | $1e-4$ | 0. | AdamW(0.9, 0.9) |
| VDVAE | 32 | 2000 | 0. | 2.2 | $1e-4$ | 0. | AdamW(0.9, 0.9) |

Table 4: Optimization settings for training baseline VDVAE and Collage VAE (ours) on dynamically binarized MNIST.

# D  Additional Results

## D.1  Super-Resolution of Collage VAE Samples

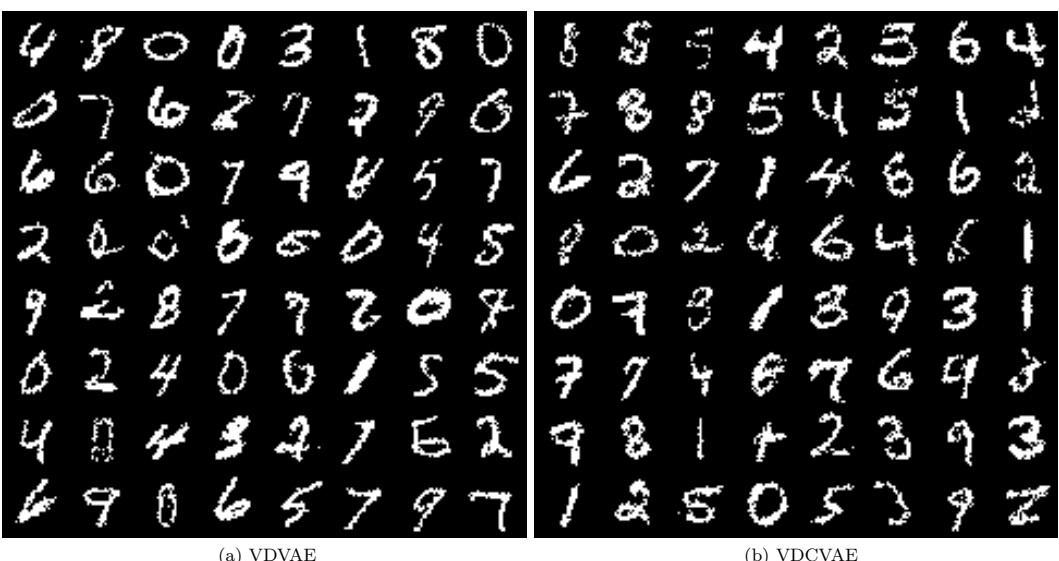

(a) VDVAE  (b) VDCVAE

Figure 10: Super-resolution factor of $1\times$ the original data resolution ($28 \times 28$ MNIST). **[Left]:** Samples from the VDVAE baseline. **[Right]:** Samples from a Collage VAE.

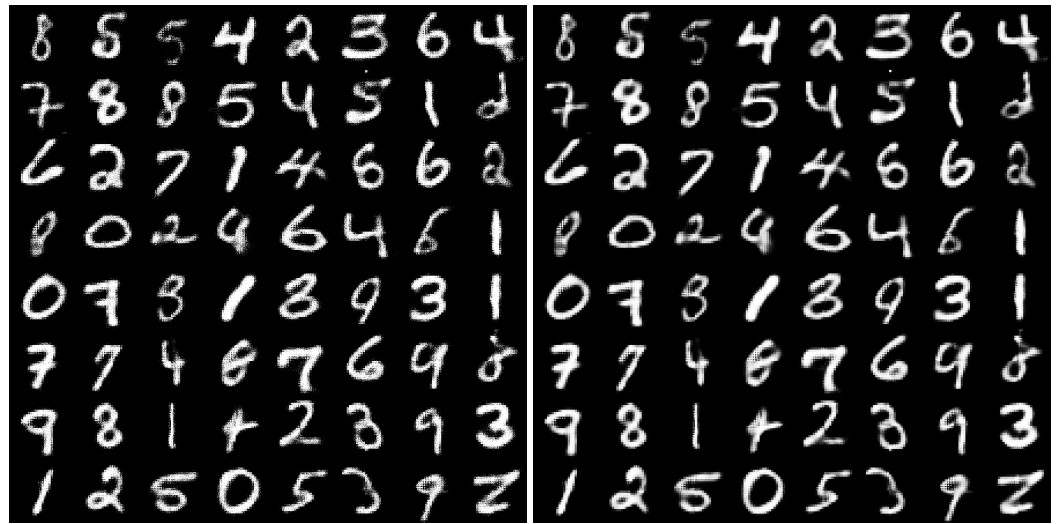

Figure 11: **[Left:]** `Collage` VAE samples with $10\times$ magnification ($280 \times 280$ resolution). **[Right:]** `Collage` VAE samples with $40\times$ magnification ($1120 \times 1120$ resolution).

### D.2 Compression

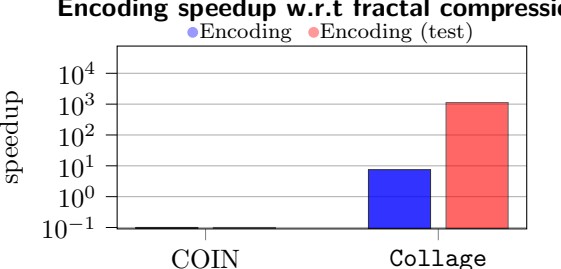

Figure 12: Wall-clock encoding speedups of Neural `Collage` compressors and COIN over a PIFS-based fractal compression implementation on GPU.

## D.3 Fractal Stylization

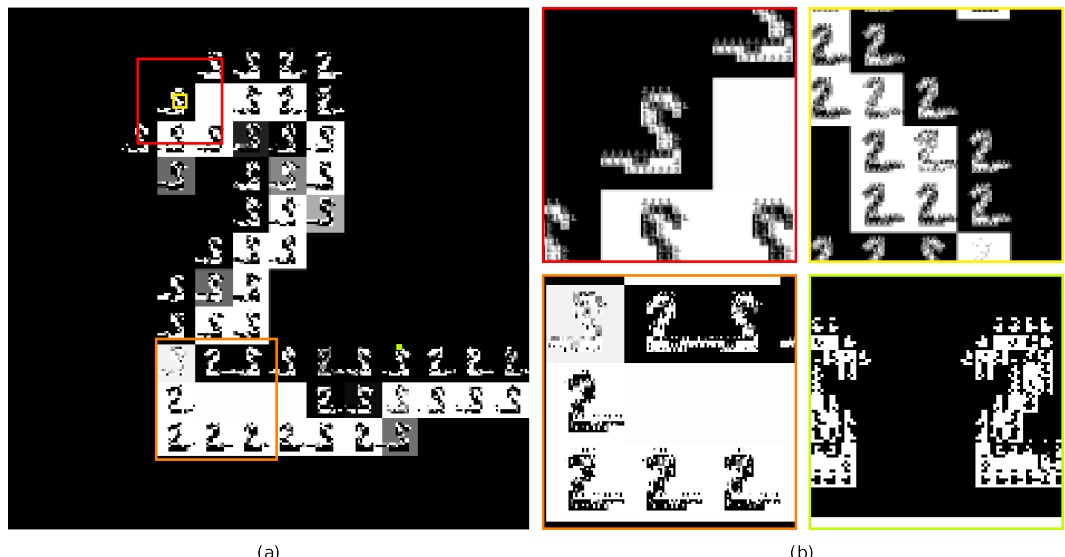

(a)                 (b)

Figure 13: Fractalized MNIST digit via a `Collage` at $500\times$ the original data resolution. Red box is a $3\times$ magnification, yellow is a $20\times$ magnification, orange is $2\times$ magnification, and lime is $80\times$ magnification. Note that the image is compressed for easier displaying within the pdf. Refer to the code repository for the full quality version.

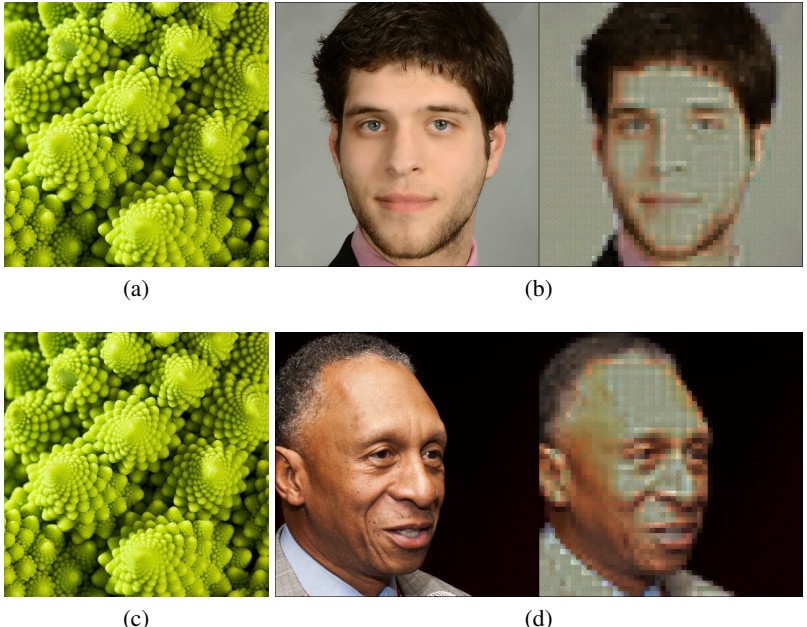

(a)                 (b)

(c)                 (d)

Figure 14: Neural `Collage` texturizer, with relative weights of $0.5$ for U (texture source) and $1.0$ for D (domain cells) in the `Collage` iteration.