# OpenReview forum: "Self-Similarity Priors: Neural Collages as Differentiable Fractal Representations"
_NeurIPS.cc/2022/Conference — NeurIPS 2022 Accept_

### Official Review · Reviewer_d8rc · 2022-07-11

**Rating:** 5
**Confidence:** 3
**Soundness:** 3 good
**Presentation:** 3 good
**Contribution:** 3 good

**Summary:**

This paper proposes a collage function which is parameterized to be able to generate variable-resolution images according to a fractal pattern.  To be able to handle datasets/distributions, this collage function is parameterized using a hypernetwork which takes the entire example as its input.  This is used to achieve improved image compression and variable-resolution image generation.

Notes:

  -Natural patterns described using "self-similarity", self-referential transformation.

  -Represent data with self-referential transform, in single forward pass.

  -Collage VAE can generalize over multiple possible image resolutions.

  -Given an image z, iterate fixed point map: z[t+1] = A(w)z[t] + b(w)

  -Constrain collage function to be a contraction.

  -We can fit this separately for each image, or use hypernet based on a full-image encoder.


**Questions:**

To what extent do datasets in machine learning really have a fractal structure?  I feel like this paper is somewhat split between completely artificial examples (like the snowflake made of snowflake images) and natural image datasets where the existence of fractal structure is non-obvious.  It would be nice if there were more datasets with a fractal structure which is somewhat organic.  Perhaps satellite images of weather patterns?

**Limitations:**

Yes I don't think there is an ethical issue, and the limitations are fairly straightforward.

**Strengths And Weaknesses:**

Strengths:

  -The ability to generate with dynamics resolutions is an interesting property.

  -The improvements on image compression are nice but the baselines all use either implicit representations, fractals, or non-neural (if I read it correctly).

  -The subject of the paper is fairly creative and seems unique.

Weakness:

-(Minor): In Figure 1, it might be more compelling to give a natural machine learning example on the top row.  Although I suppose the fact that the fractal structure is exact is perhaps worth something.

  -(Minor): In Figure 5, the red/blue outlines around the sub-images should be much thicker.  Many readers will miss it currently.

  -Constructing the parameters of the collage function F may be a lot to ask an encoder network to do.  If I understand it correctly, this means compressing the entire object's information into a single vector (or a small set of vectors, independent of the size of the object) which is used to parameterize F?

  -The experimental strategy is a bit odd to me.  It seems like the best result to show would be that the method improves the generative modeling of data with a clear fractal structure (either hand-designed or natural).  Then, it would be interesting to show that other deep learning methods fail to generalize over this fractal structure (which I suppose they are limited due to having a fixed resolution, but could be interesting to study over some range of resolutions).

---

> ### Author Response · Authors · 2022-07-29
> **Discussion with Reviewer d8rc**
>
> Thank you for the review and great questions. We are glad to see you found the soundness, presentation and contribution good!
>
> ```
>  Q: Constructing the parameters of the collage function F may be a lot to ask the encoder to do.  If I understand it correctly, this means compressing the entire object's information into a single vector [...] which is used to parameterize F?
>  ```
>
> That's spot on! The encoder is compressing each image into the corresponding vector of Collage iteration parameters $\omega$. The size of this vector does not depend on the resolution of the image, but depends on the source and range partitioning schemes. For tiling partitions, smaller tiles correspond to a larger vector $\omega$, as each additional range tile requires two more elements in $\omega$ for each source tile. However, smaller tiles also yield more accurate results. We have included a small example in the [gist](https://gist.github.com/anonymous-conf-sub/dd01870df49a5fdc65d3a99a41abed30) (see `decoding_patch_4` and `decoding_patch_8`. We decode the same test image using  x  and  x  range tiles, without using auxiliary domains.).
>
> The fractal data encoding problem is similar to autoencoding which we know to be solvable by neural networks. We agree that it is perhaps surprising (and in our opinion, also impressive!) that a Neural Collage encoder can learn to map each image to its Collage parameters in a single forward pass. We believe this to be a major contribution of the paper.
>
> ```
> The improvements on image compression are nice but the baselines all use either implicit representations, fractals, or non-neural (if I read it correctly).
> ```
> We compare against closely related compression methods: fractal compression (with different domain augmentations), COIN (which is a deep learning-based compression method), and lossy compression with block-DCT.
>
> ```
>  Q: It seems like the best result to show would be that the method improves the generative modeling of data with a clear fractal structure (either hand-designed or natural) [...] to show that other deep learning methods fail to generalize over this fractal structure (which I suppose they are limited due to having a fixed resolution, but could be interesting to study over some range of resolutions).
> ```
>
> Most datasets fall somewhere on the scale of "self-similarity free" to "fractal", i.e. perfectly defined at any resolution by an iterated map. Your suggestion is sound, and it is exactly what we have attempted with the Collage VAE experiment. Although MNIST digits clearly do not possess any intrinsic fractal structure, it can be seen how they are (partially) self-similar at a "patch-level". Collage VAEs are able to detect and exploit this self-similarity, whereas regular VAEs as you mention cannot due to being limited to a fixed resolution. Comparisons of samples at different resolutions are shown in Figure 5 of the main text, and Figure 10 and 11 in Appendix D.
>
>
> ```
> Q: To what extent do datasets in machine learning really have a fractal structure? I feel like this paper is somewhat split between completely artificial examples (like the snowflake made of snowflake images) and natural image datasets where the existence of fractal structure is non-obvious. It would be nice if there were more datasets with a fractal structure which is somewhat organic. Perhaps satellite images of weather patterns?
> ```
>
> Great suggestion! In fact, our original application for this work was aerial imagery of farmlands or other locations with repeated patterns - also the reason behind our choice to utilize high-resolution aerial images for the compression task with Neural Collages in Section 4.2.
> We would like to highlight a subtle point about self-similarity. Consider for example a simulated landscape generated using a fractal algorithm (or even a "classical" fractal such as the Mandelbrot set), and then images of it taken from different perspectives. "Fractalness" of the object does not necessarily imply that the resulting perspectives would be well-suited to self-similarity-based methods such as Neural Collages. In particular, Neural Collages exploit self-similarity at a tile level (in pixel space!), which can be present even when the object in the image is not fractal.
>
> We strongly agree that there are many stimulating questions at the intersection of self-similarity methods and deep learning, certainly far too many to answer in a single paper. Our goal with this work and method is to provide compelling evidence that Neural Collages and self-similarity can have impact on various deep learning applications - we hope to have convinced you of the same!

---

> > ### Author Response · Authors · 2022-08-09
> > **Request for feedback**
> >
> > As the author-reviewer discussion phase is coming to an end, please let us know if there are further questions or concerns to address.

---

### Official Review · Reviewer_zn4B · 2022-07-12

**Rating:** 5
**Confidence:** 2
**Soundness:** 2 fair
**Presentation:** 1 poor
**Contribution:** 3 good

**Summary:**

This paper studies using learning algorithms to improve fractal compression. The fractal compression leverages self-similarities to compress the data, which happens in many common signals, such as images. However, the expressiveness of the compression is constrained by the model families it uses. Traditional methods use affine transformations, which is limited. This paper consider a learning-based method, which they call Collage. They also propose a hypernetwork to tackle the potential optimization difficulties.


**Questions:**

- It seems we need some conditions to ensure the convergence when solving the optimization problem. However, I don’t see it to be explicitly imposed in the algorithm design. Do I miss it in the paper anywhere?

- When solving the optimization problem below line 228, usually how many iterations do we need to get z*? And if the iteration is a lot, would that create any issues? (i.e. RNN with many time stamps.)

- There are many algorithms proposed to use a single feedforward pass of a neural network to solve different optimization problems. However, many problems are not really solved well by neural networks. How about the case of the hypernetwork here?



**Ethics Review Area:**

["I don’t know"]

**Limitations:**

No. The authors mention in Appedix B.2, however, there is only Appendix B.1.  Also, it would be great to discuss the limitation too. As it's a new attempt on this compression method, I tend to believe there are still rooms to improve.

**Strengths And Weaknesses:**

# Strong

A new learning- based method to improve a classic compression algorithm with significant performance improvements as shown numerically and visually in the experiments.

# Weak

- The paper is not easy to follow to be honest as a paper in a machine learning venue. Although the authors spend quite an amount of effort describing fractal compression in Section 2, and covers some theoretical results. In the end, I have to read other papers first to understand the idea.  I would suggest the author to provide more illustrative examples to guide readers not in compression background to understand the task and shortens some mathematical properties by moving to the appendix, as they are proven by previous literature and not really mentioned in the main algorithm design.  I want to note that I’m criticizing the paper working on applications from other domains. It would be better if the authors make the problem more clear to the readers as it is in a ML venue not in the signal processing venue.   This comment might be biased as I’m not an expert in this domain.

- It would be nice to provide more methodology details. For example, what is J here in the final equation which depends on d_H, also what’s the hypernetwork architecture and the output .. etc. Providing more algorithm details would greatly help reader to capture the idea of the proposed method.

- The related works are shortened too much. It would be great if the authors provide a better overview of the recent progress of learning-based compression methods in the related works. For example, in addition to the very first NeRF paper, there are many great follow-up works worth mentioning.  A suggestion is the same as above by moving some non-necessary results from the prior works to the appendix to gain more space.

---

> ### Author Response · Authors · 2022-07-29
> **Discussion with Reviewer zn4B [1/2]**
>
> Thank you for the review and the interesting questions. Hopefully our answers below will clarify some of the doubts!
>
> ```
> The paper is not easy to follow to be honest as a paper in a machine learning venue
> ```
> We appreciate the honest feedback! Section 2 represents our best attempt at presenting essential background on fractal compression required for Neural Collages. It would be very helpful if you could point us to passages that were less intuitive to follow.
>
> Here are some concrete steps we are planning - we think these changes will improve the presentation and readability of this work:
>
> * An additional toy example before Section 2. The example will introduce the reader to the (highly non-intuitive) idea of representing data as the parameters of an iterated map. Please see `intro_example` and `intro_example_anim` in the [anonymous gist](https://gist.github.com/anonymous-conf-sub/dd01870df49a5fdc65d3a99a41abed30).
> * Shorten 2.1 and 2.2, moving some technicalities to the appendix.
> * Add Python / PyTorch pseudocode for a single Collage iteration on images, further contextualizing Figure 2. This, in combination with eq (3.1) and Figure 2, should give more options for the reader to grasp the concept of Neural Collages, suited to their background and preferences.
> * Improve readability of some Figures. In particular, making the red / blue outlines in Figure 5 easier to see.
>
> ```
> It would be nice to provide more methodology details. For example, what is J [...] what’s the hypernetwork architecture and the output
> ```
> Absolutely! $J$ in equation (3.3) is the loss function that depends on the task. In the compression experiments, we use a reconstruction loss (MSE) between the image $z^*$ decoded as the fixed-point of a Collage iteration, and the input image $x$. Note how we indicate the dependence of the fixed-point $z^*$ on the Neural Collage encoder parameters $\theta$: indeed, the fixed-point depends on Collage parameters $\omega$, which are produced as the output of the encoder.
>
> The hypernetwork (Neural Collage encoder $\mathcal{E}_{\theta}$) takes as input batches of images $x$ and is trained to output a set of Collage iteration parameters $\omega$, such that $J(z^*, x)$ is minimized.
>
> The question on how to design Neural Collage encoder architectures is particularly interesting. Please see how extended response to Reviewer 4iu4 on the topic. In short: once partitioning schemes are chosen, Neural Collage encoders are trained map input images to a vector of fixed dimension, not unlike image classification. We find that all performant architectures for image classification (ConvMixer, ViT, MLPMixer, among others) are capable of being performant encoders.
>
> ```
> The related works are shortened too much. It would be great if the authors provide a better overview
> ```
> We agree. There is an enormous quantity of inspiring work at the intersection of deep learning, compression and implicit representations. We chose to highlight those that (despite not being always methodologically related) inspired us most while developing Neural Collages. For example: SIREN (Sitzmann et al) and COIN (Dupont et al),
>
> We plan to several additional references to follow-up work:
>
> * COIN++: Data Agnostic Neural Compression
> * NeRF in the Wild: Neural Radiance Fields for Unconstrained Photo Collections
> * Neural Sparse Voxel Fields
>
> Further suggestions are very welcome!
>
> ```
> It seems we need some conditions to ensure the convergence when solving the optimization problem. [...] Do I miss it in the paper anywhere?
> ```
> That's correct! Collage iterations need to be contractive in order for the fixed-point $z^*$ to exist. As our Collage consists of affine maps applied to pixels of a patch, a sufficient condition for convergence is $|a|<1$, where $a$ is the "scaling" factor in $z_{k+1} = a z_{k} + b$
>
>
> We enforce sufficient conditions by applying a tanh activation as the last step of a Neural Collage encoder, effectively bounding $a$ in the interval $(-1, 1)$ for each map in the Collage iteration. In our experience, the encoder never outputs $a$ exactly equal to either $-1$ or $1$. There is a brief mention to tanh in the Appendix (C.4, Neural Collages) and main text (Section 4.2, Code length of a Collage), in the context of bits-per-dim computation. We will highlight these important details further in the Section 3 of the main text, during the presentation of the core Neural Collage algorithm.

---

> > ### Author Response · Authors · 2022-07-29
> > **Discussion with Reviewer zn4B [2/2]**
> >
> > ```
> > When solving the optimization problem below line 228, usually how many iterations do we need to get z*? And if the iteration is a lot, would that create any issues? (i.e. RNN with many time stamps.)
> > ```
> >
> > Learned collage iterations are in our experience "very" contractive i.e. it takes few steps to reach $z^*$. For example, we use only $5$ iterations for Collage VAEs. As such, there are no notable numerical error propagation issues, which as the reviewer points out would definitely occur if longer rollouts were required.
> >
> > As an interesting sidenote, we note that one could use the implicit function theorem to compute the gradients with respect to the collage parameters at the fixed point $z^*$, similarly to what is done for example in Deep Equilibrium Models or Differentiable Multiple Shooting Layers.
> >
> > ```
> > There are many algorithms proposed to use a single feedforward pass of a neural network to solve different optimization problems. However, many problems are not really solved well by neural networks. How about the case of the hypernetwork here?
> > ```
> >
> > It is true that neural networks used to amortize the cost of finding the solution of an optimization problem do not always work particularly well. Luckily, the fractal data encoding problem for Neural Collages is well-posed: all the information required by the encoder to produce Collage iteration parameters $\omega$ is fully contained in the input image, and no approximation or relaxation is required as every step is differentiable.
> >
> > The problem solved by Neural Collage encoders is in spirit of similar complexity to the autoencoding problem, which we know to be solvable by neural nets: the encoder searches for a lower-dimensional representation such that a decoder can reconstruct the original data. In the case of Neural Collages, the encoder is learned, while the decoder has a specific structure, the Collage iteration. This structure provides the overall method with a variety of useful properties, such as the ability to decode at different resolutions.
> >
> > ```
> > The authors mention in Appedix B.2, however, there is only Appendix B.1. Also, it would be great to discuss the limitation too. As it's a new attempt on this compression method, I tend to believe there are still rooms to improve.
> > ```
> >
> > Thank you for noticing! An additional section will be added to the Appendix, discussing limitations and areas open to further investigation.
> >
> > To be specific, here are some limitations and avenues of improvement for Neural Collages we think would be fruitful:
> >
> > * Improvements to the partitioning schemes. We have so far studied tiling source and range partitions for images. More flexible partitioning schemes would allow Neural Collages to leverage self-similarity across domains of different shapes.
> > * Additional data modalities: other types of data, such as audio or graphs, also possess degrees of self-similarity.
> > * Analyzing the type of detail introduced in images by decoding Neural Collages at higher resolutions. We do not yet fully understand how to control detail in upsampled images through a particular choice of domain augmentations and partitioning schemes.

---

> > > ### Author Response · Authors · 2022-08-09
> > > **Request for feedback**
> > >
> > > Thank you for raising your score and acknowledging the rebuttal. Please let us know if there are further questions or concerns to address.

---

### Official Review · Reviewer_4iU4 · 2022-07-12

**Rating:** 6
**Confidence:** 2
**Soundness:** 3 good
**Presentation:** 2 fair
**Contribution:** 4 excellent

**Summary:**

Fractal compression represents data (for example an image) as the fixed point of a parameterized operator. Encoding the image requires finding the parameters of the operator (typically done via an explicit search algorithm), while decoding the image requires iterating the operator to recover the fixed point.

This paper proposes neural collages, which as in fractal compression represent data as the fixed point of a parameterized map, but uses hypernetworks to amortize the cost of finding these parameters.

The paper introduces the method, and uses it as the decoder for deep generative models, for image compression, and to generate fractal art.

**Questions:**

- What was the architecture used for the hypernetwork? Is there any intuition or heuristics for how to design that architecture? I don't have any intuition for what one might expect to work there.
- Could you add a little more detail about the auxiliary domains? How much do those affect the overall performance?

**Limitations:**

yes

**Strengths And Weaknesses:**

- The core idea is original, and an interesting way of leveraging neural networks for fractal image compression.
- The applications nicely demonstrate the wide applicability of the method.
- The presentation of the method could be clearer, with more intuition to accompany the idea, and some clearer distinction between what is prior background vs. new ideas.

---

> ### Author Response · Authors · 2022-07-29
> **Discussion with Reviewer 4iU4**
>
> Thank you for the review and questions. We are glad to see the Neural Collage contribution described as excellent!
> ```
> Q: What was the architecture used for the hypernetwork? Is there any intuition or heuristics
> ```
> Interesting question. The size of collage parameter vector $\omega$ does not depend on image resolution, but instead depends on source and range partitioning schemes. For tiling partitions, smaller tiles correspond to a larger vector $\omega$, as each additional range tile requires two more elements in $\omega$ for each source tile. Smaller tiles tend to yield more accurate results when decoding through a Collage (see for example `decoding_patch_4` and `decoding_patch_8` in this [anonymous gist](https://gist.github.com/anonymous-conf-sub/dd01870df49a5fdc65d3a99a41abed30). We decode the same test image using $4$ x $4$ and $8$ x $8$ range tiles, without using auxiliary domains.).
>
> As an example, $128$ x $128$ images with $64$ x $64$ domain tiles and $8$ x $8$ range tiles require a collage parameter vector with $2048$ elements ($16$ x $16 = 256$ range tiles, $4$ domain tiles, $2$ parameters for every combination).
>
> Once partitioning schemes are chosen, Neural Collage encoders are trained map input images to a vector of fixed dimension, not unlike image classification. We find that all performant architectures for image classification (ConvMixer, ViT, MLPMixer, among others) are capable of being performant encoders. In most of our experiments - as well as in the example above - the size of the collage parameter vector is in the thousands, corresponding in size to logit vectors of ImageNet classifiers. In our aerial image compression experiments, we use an architecture similar to ConvMixers: we first "patchify" the input image, splitting it into a vector of embeddings for each range tile, then pass these through a block of pointwise convolutions ($2$ pointwise convolutions in each, with an activation in between) and a block of spatial mixing layers through group convolutions. We repeat channel and spatial mixing $4$ times.
>
> To ensure convergence of the Collage iteration given collage parameters, we apply a tanh as the last step of the encoder. Since the maps are affine, costraining $|a|<1$, where $a$ is the "scaling" factor in each affine map $z_{k+1} = a z_{k} + b$ is sufficient.
>
> ```
> Q: Could you add a little more detail about the auxiliary domains? How much do those affect the overall performance?
> ```
>
> We find that introducing auxiliary domains consistently improves performance in compression and generation tasks. Since real data is never perfectly self-similar, introducing the ability to "extract" common patterns across images as auxiliary domains for the Collage iterations makes the method more robust to a wider range of datasets compared to standard fractal compression. In the aerial image compression experiment, removing auxiliary patches reduces PSNR, partially closing the gap with vanilla fractal compresson (~ -1 PSNR at high bpp).
>
> Interestingly, we experimented with different methods to learn auxiliary domains in generation and compression. In particular, we attempted (1) direct parametrization and optimization of auxiliary tiles via gradient descent and (2) auxiliary tiles as latent variables in the Collage VAE, with data-independent priors and variational posteriors parametrized by small neural networks. We found (1) to yield more detailed reconstructions and samples.
>
> ```
> The presentation of the method could be clearer, with more intuition to accompany the idea
> ```
>
> Thank you for the feedback! With the exception of "A learning perspective of fractal data encoding", Section 2 is meant to contain background and summary of prior work on fractal compression we think is necessary to grasp the core idea behind Neural Collages. From Section 3, the paper contains new ideas related to the proposed Neural Collage.
>
> Here are some concrete steps we are planning that we think will improve the presentation and readability of the work (we would be happy to take further suggestions into account):
>
> * An additional toy example before Section 2. The example will introduce the reader to the (highly non-intuitive) idea of representing data as the parameters of an iterated map. Please see the `intro_example` and `intro_example_anim` in the [anonymous gist](https://gist.github.com/anonymous-conf-sub/dd01870df49a5fdc65d3a99a41abed30).
> * Shorten 2.1 and 2.2, moving some technicalities to the appendix.
> * Add Python / PyTorch pseudocode for a single Collage iteration on images (Collage pseudocode in the [gist](https://gist.github.com/anonymous-conf-sub/dd01870df49a5fdc65d3a99a41abed30)), further contextualizing Figure 2. This, in combination with eq (3.1) and Figure 2, should clarify the core steps behind Neural Collages, improving accessibility to readers from a wider range of backgrounds.
> * Improve readability of some Figures. In particular, making the red blue outlines in Figure 5 easier to see.

---

> > ### Author Response · Authors · 2022-08-09
> > **Request for feedback**
> >
> > As the author-reviewer discussion phase is coming to an end, please let us know if there are further questions or concerns to address.

---

### Author Response · Authors · 2022-08-02
**Summary of responses**

We thank all reviewers for their detailed and thoughtful comments. We are glad to receive unanimous positive comments on our contribution and evaluation: "excellent", "original" (Reviewer 4iU4), "significant performance improvements" (Reviewer zn4B) "fairly creative, unique" (Reviewer d8rc). We focus on improving the clarity of the background section (Section 2), as pointed out in the reviews.

Our planned changes to improve the readability of our manuscript include:

* An additional toy example before Section 2. The example will introduce the reader to the (highly non-intuitive) idea of representing data as the parameters of an iterated map. Please see the `intro_example` and `intro_example_anim` in the [gist](https://gist.github.com/anonymous-conf-sub/dd01870df49a5fdc65d3a99a41abed30).
* Shorten 2.1 and 2.2, moving some technicalities to the appendix.
* Add Python / PyTorch pseudocode for a single Collage iteration on images (Collage pseudocode in the [gist](https://gist.github.com/anonymous-conf-sub/dd01870df49a5fdc65d3a99a41abed30)), further contextualizing Figure 2. This, in combination with eq (3.1) and Figure 2, should clarify the core steps behind Neural Collages, improving accessibility to readers from a wider range of backgrounds.

We believe your comments helped us greatly improve the manuscript. We are happy to answer any additional questions should they arise.

---

### Meta-Review · Area_Chair_s1rd · 2022-08-23

**Recommendation:** Accept
**Confidence:** Certain

**Metareview:**

This paper introduces neural collages, which are operators that aim to represent an image via parameters to a self-referential transformation. The parameters for a given are predicted by a hypernetwork. The approach is mainly evaluated on image compression, but applications to generative modeling are considered. While reviewers had some trouble reviewing the submission because it falls out of their area of expertise, they generally agreed that the approach was novel, interesting, and had compelling applications.

**Award:**

No

---

### Decision · Program_Chairs · 2022-09-14

Accept